# FastAvatar: Towards Unified and Fast 3D Avatar Reconstruction with Large Gaussian Reconstruction Transformers

**Yue Wu**[1,2] **Xuanhong Chen**[3*] **Yufan Wu**[3] **Wen Li**[4] **Yuxi Lu**[1] **Kairui Feng**[1,2]

Tongji University[1]   Shanghai Innovation Institute[2]   Shanghai Jiao Tong University[3]   AKool[4]

yuewu@tongji.edu.cn, chenxuanhong@sjtu.edu.cn, kelvinfkr@tongji.edu.cn

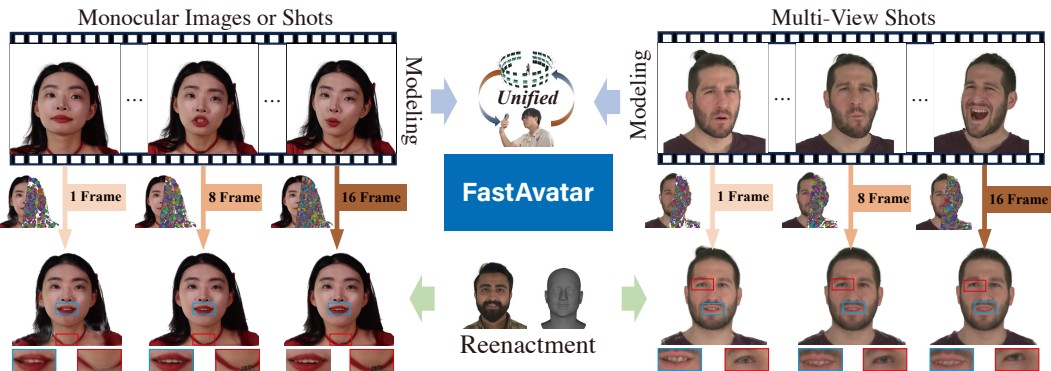

Figure 1: Unlike existing 3D Avatar methods that can only process fixed-length data, FastAvatar achieves incremental reconstruction. It can strike a good balance between modeling quality and inference speed based on available data volume, delivering high-quality models with sufficient data while providing viable reconstruction results at high speed even with limited data.

## Abstract

Despite significant progress in 3D avatar reconstruction, it still faces challenges such as high time complexity, sensitivity to data quality, and low data utilization. We propose **FastAvatar**, a feedforward 3D avatar framework capable of flexibly leveraging diverse daily recordings (e.g., a single image, multi-view observations, or monocular video) to reconstruct a high-quality 3D Gaussian Splatting (3DGS) model within seconds, using only a single unified model. The core of FastAvatar is a Large Gaussian Reconstruction Transformer (LGRT) featuring three key designs: First, a 3DGS transformer aggregating multi-frame cues while injecting initial 3D prompt to predict the corresponding registered canonical 3DGS representations; Second, multi-granular guidance encoding (camera pose, expression coefficient, head pose) mitigating animation-induced misalignment for variable-length inputs; Third, incremental Gaussian aggregation via landmark tracking and sliced fusion losses. Integrating these features, FastAvatar enables incremental reconstruction, i.e., improving quality with more observations without wasting input data as in previous works. This yields a quality-speed-tunable paradigm for highly usable 3D avatar modeling. Extensive experiments show that FastAvatar has a higher quality and highly competitive speed compared to existing methods. Code and models are available at https://github.com/TyrionWuYue/FastAvatar.

## 1 Introduction

Creating photorealistic 3D avatar reconstruction is one of the fundamental problems in computer vision and graphics. Contemporary methods Kirschstein et al. (2025); He et al. (2025); Chen et al.

---

*Corresponding authors

(2024c); Qian et al. (2024a); Pan et al. (2024); Wen et al. (2024); Qian et al. (2024b); Jiang et al. (2024); Hu et al. (2024); Qiu et al. (2025) for 3D avatars have made significant advancements in 3D representation and modeling quality. However, these approaches commonly suffer from drawbacks such as data sensitivity (e.g., requiring richly expressive data), high time complexity, and low data utilization efficiency. These issues, pose severe challenges to the low-cost application of 3D avatars.

Three factors hinder existing 3D avatar methods from addressing the aforementioned challenges: 1) *Inability to Leverage Prior Knowledge*: Although contemporary 3D avatars have widely adopted efficient representations like 3DGS Kerbl et al. (2023), they still primarily rely on per-scene optimization. This approach fails to utilize "experience" from similar scenes, preventing the acquisition of good initial values to accelerate optimization. Furthermore, since all model information stems solely from the input observations, missing data cannot be reconstructed, resulting in a heavy dependence on complete 3D observations. Daily captures, however, often contain significant information gaps. 2) *Low Accuracy in Observation Alignment*: 3D avatar methods typically depend entirely on parametric proxy models (e.g., 3DMM/FLAME Blanz & Vetter (1999); Li et al. (2017)) for coarse view alignment. The precision of this alignment is critical for effective modeling; For instance, GaussianAvatars Qian et al. (2024a) even requires the proxy model to provide detailed meshes for hair. However, these parametric models are susceptible to limitations in representational capacity (e.g., blendshapes from 3D scan databases), lighting conditions, and data quality, often failing to produce highly accurate proxy 3D models. Using this proxy without refinement leads to poor robustness, hindering unified adaptation to diverse data sources (e.g., light fields, smartphones, DSLR cameras). 3) *Inadequate Handling of Variable-Length Data*: Optimization-based 3D avatar methods typically require input data of a minimum specific length (typically at least 30 seconds at 25fps). Insufficient data often leads to modeling failure, resulting in severely limited capability to process few-shot data (e.g., 1 frame, 4 frames). Meanwhile, recently proposed feedforward-style methods Kirschstein et al. (2025); He et al. (2025) are usually designed for fixed-length inputs for training convenience. For instance, LAM He et al. (2025) can only process single-frame input, and Avat3r Kirschstein et al. (2025) is fixed to handling exactly 4 frames. However, real-world data can consist of any arbitrary number of frames. The inability to process inputs of variable lengths will result in wasted valuable observation data, consequently limiting modeling quality.

To pursue data-efficient, high-quality, and fast 3D avatar reconstruction, we propose *FastAvatar*. It enables direct feedforward reconstruction of animatable avatars within seconds from arbitrary-length input frames and can incrementally leverage additional observation data. The core of FastAvatar is a **L**arge **G**aussian **R**econstruction **T**ransformer (LGRT). It can align and aggregate variable-length facial inputs based on head pose and camera pose, then generate high-quality Gaussian model groups using coarse 3D positional prompts. Finally, these groups can be flexibly fused into a single 3DGS avatar model according to quality requirements and computational resources. Notably, compared to LAM He et al. (2025) and Avat3r Kirschstein et al. (2025), FastAvatar handles variable-length observation data with greater model flexibility and higher data utilization efficiency. Unlike VGGT Wang et al. (2025a), FastAvatar is capable of directly generating 3DGS avatars and can achieve granular 3D model aggregation. Benefiting from the successful architecture of VGGT, LGRT is designed as a variant of the VGGT structure. We replace the unstable Dense Prediction Transformer (DPT) Ranftl et al. (2021) with an MLP that directly predicts canonical 3DGS models, while adopting 3D parametric model (e.g., FLAME) mesh vertices as positional prompts for the output. These improvements maximize adaptability for the prediction of the 3DGS avatar. Instead of relying solely on single camera pose encoding, 3D avatar reconstruction demands higher requirements for input data alignment. Therefore, we additionally incorporate expression coefficients and head pose as positional encoding for input observations, enabling more precise cross-frame information aggregation. Critically, we propose a landmark tracking loss and sliced fusion loss to efficiently supervise the model for enhanced aggregation accuracy while enabling incremental 3DGS models fusion. Integrating these key designs, our model pioneers incremental 3D avatar reconstruction, meaning it can continuously ingest new observational data to progressively refine modeling quality.

Extensive experiments demonstrate that our model achieves highly competitive 3D reconstruction quality compared to state-of-the-art methods. It uniquely accomplishes incremental 3D avatar reconstruction, currently unattainable by existing approaches, and holds promise for delivering favorable solutions in the quality-speed trade-off paradigm.

## 2 RELATED WORK

### 2.1 3D-BASED HEAD AVATAR RECONSTRUCTION

FLAME-based Li et al. (2017); Feng et al. (2021); Daněček et al. (2022); Ma et al. (2024); Cudeiro et al. (2019) techniques utilize a parametric model in head reconstruction, allowing for effective expression control but struggle to represent details (e.g., eyes, teeth) and limited to single-view. Since neural radiance fields have demonstrated strong ability to synthesis photo-realistic images, some method Zielonka et al. (2023); Shao et al. (2023); Athar et al. (2023); Müller et al. (2022) have adopted NeRF Mildenhall et al. (2021) for head reconstruction, which perform higher fidelity, particularly in modeling fine-scale details like hair. However, NeRF-based approaches Athar et al. (2022); Guo et al. (2021); Liu et al. (2022) often suffer from a significant issue with head rendering speed limitations and extensive training data. Recently, 3DGS Kerbl et al. (2023) has demonstrated superior performance surpassing NeRF in both novel view synthesis quality and rendering speed. Approaches Qian et al. (2024a); Chen et al. (2024c); Xu et al. (2024); Wang et al. (2025b); Wu et al. (2025) generate photorealistic head avatars that allow full control over expressions and poses using multi-view videos. Another line of research places explicit emphasis on identity preservation Gerogiannis et al. (2025); Zheng et al. (2025); Zielonka et al. (2025). Despite 3DGS's impressive performance, it requires multi-frame data for identity-specific training and lacks flexibility, necessitating separate models for single-view and multi-view scenarios. In contrast, our FastAvatar achieves ultra-fast 3D head avatar reconstruction with a unified model.

### 2.2 FEED-FORWARD RECONSTRUCTION MODEL

Traditionally, 3D reconstruction and view synthesis, depending on optimization-based approaches such as Structure-from-Motion Schonberger & Frahm (2016) and Multi-View Stereo Schönberger et al. (2016), are often computationally intensive, slow to converge, and reliant on precisely calibrated dataset, limiting their applications in real-world scenarios. Recently, series of research Wang et al. (2024); Chen et al. (2024b); Liu et al. (2024); Ye et al. (2024); Hong et al. (2023); Charatan et al. (2024); Szymanowicz et al. (2024); Tang et al. (2024); Jin et al. (2024); Zhang et al. (2024a); Jiang et al. (2025) initiate a new research paradigm termed Feed-forward 3D reconstruction model. DUSt3R Wang et al. (2024) introduces a method for dense and unconstrained stereo 3D reconstruction, operating without prior camera calibration or viewpoint poses. VGGT Wang et al. (2025a) uses a large feed-forward transformer to effectively predict all key 3D attributes from a single image or multiple images. While feed-forward networks excel in generic 3D reconstruction, their application to 3D head avatar reconstruction is still nascent and warrants systematic exploration. LAM He et al. (2025) introduces a feed-forward framework that reconstructs an animatable gaussian head from a single image, allowing animation and rendering without additional post-processing. Avat3r Kirschstein et al. (2025) regresses animatable 3D head avatar from just a few input images, reducing compute requirements during inference. A key challenge in Feed-forward Head Reconstruction Model is the absence of a unified framework to handle diverse real-world inputs, including monocular videos, sparse multi-view captures. To address this gap, we propose a VGGT-style transformer architecture to jointly process different observation resource, achieving state-of-the-art performance.

## 3 METHODOLOGY

Daily observations are diverse and varied, such as single selfies, multi-angle selfies, video vlogs, etc. In summary, they are variable-length. Existing optimization-based 3D Avatar methods Kirschstein et al. (2025); Chen et al. (2024c); Qian et al. (2024a) cannot effectively handle overly short data (typically a single image). FastAvatar was specifically designed to address this scenario.

### 3.1 PROBLEM DEFINITION AND NOTATION

FastAvatar $\mathcal{G}(\cdot)$ is a feed-forward avatar reconstruction framework designed to take any number of input observations and output a high-quality animatable 3DGS avatar:

$$\mathcal{A} \leftarrow \mathcal{G}(I, \pi, z_{exp}, z_{pose}), \tag{1}$$

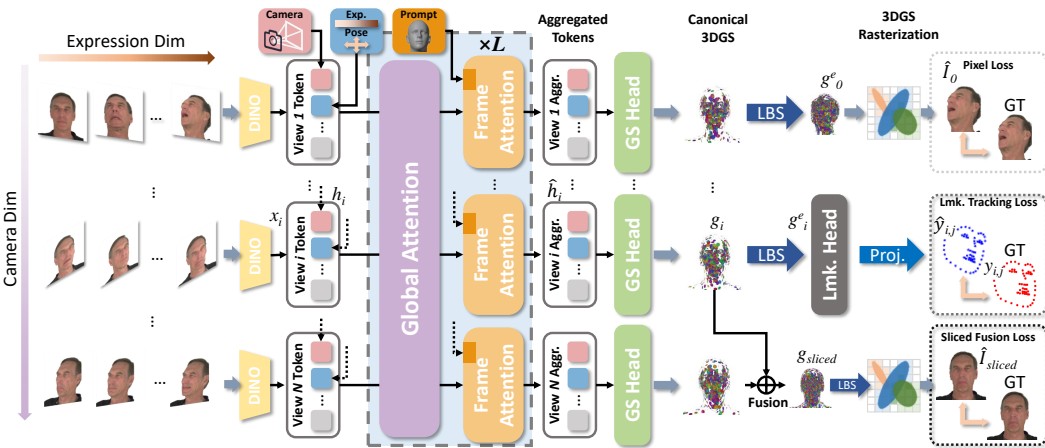

Figure 2: The core of FastAvatar is a Large Gaussian Reconstruction Transformer (LGRT), which can flexibly process input data with varying expressions, poses, and camera angles, aggregating them into a high-precision 3DGS avatar model. This capability is enabled by several key designs: the interleaving of global attention and frame attention to register complex input data while encoding 3D positional prompts; multi-granular positional information encoding; and the use of landmark tracking loss and sliced fusion loss, allowing the model to smoothly and incrementally fuse additional input data.

where $(I_i)_{i=1}^N$ denotes an unordered sequence of $N$ RGB observations, with each $I_i \in \mathbb{R}^{3 \times H \times W}$. $N$ does not exceed the maximum GPU capacity, typically $1 \sim 16$ frames. The corresponding facial expression and head pose are represented by $z_{exp}$ and $z_{pose}$, respectively. $\pi$ denotes the camera parameters and poses. The output 3D Gaussian avatar representation $g$, including color $c \in \mathbb{R}^3$, opacity $o \in \mathbb{R}$, per-axis scale factors $s \in \mathbb{R}$, rotation $R \in \mathbb{R}^4$, importance score $m \in \mathbb{R}$, and points offest $O \in \mathbb{R}^3$, can be driven by any desired expression and pose from an arbitrary viewpoint using a differentiable rasterizer $\Psi$ Kerbl et al. (2023).

## 3.2 Large Gaussian Reconstruction Transformer

The core of FastAvatar is a Large Gaussian Reconstruction Transformer (LGRT). The LGRT is redesigned specifically for 3D avatar tasks, which demand finer granularity compared to SLAM-based environmental reconstruction. Moreover, the human subjects captured for 3D avatars cannot remain perfectly static and often exhibit rich dynamic characteristics (i.e., expressions, poses, etc.). The LGRT comprises 6 stages: facial feature extraction, face encoding, face aggregation and registration, 3DGS attribute generation, canonical 3DGS model fusion, and 3DGS rasterization.

**Face Encoding** FastAvatar encodes each face observation $I_i$ to a set of token $x_i$ through DI-NOv2 Oquab et al. (2023). These tokens vary from head poses, facial expressions to camera poses, and undifferentiated aggregation would lead to over-smoothing and aliasing effects. Therefore, FastAvatar introduces three critical encodings to label distinct facial tokens, facilitating subsequent aggregation. This process can be formulated as:

$$h_i = \mathcal{U}\left(x_i, \text{MLP}([\pi_i, z_i^{exp}, z_i^{pose}])\right),  \tag{2}$$

where $\mathcal{U}(\cdot)$ denotes concatenation along the dimensional axis. $h_t$ denotes the encoded face tokens. $\pi_i$, $z_i^{pose}$, and $z_i^{exp}$ represent the camera pose, head pose, and expression coefficients of $x_i$ respectively. These are processed through a lightweight MLP layer for feature alignment and dimensionality alignment. We obtain rough initial estimates of the three parameters through multi-view FLAME tracking.

**Face Aggregation and Registration** The core enabling component for constructing a dense 3D avatar from variable-length input data lies in the aggregation and registration of face tokens. The purpose of aggregation is to extract intra-token features while incorporating initialized 3D positional

prompts. These positional prompts provide the LGRT with initial 3DGS positions, thereby accelerating 3D reconstruction. As illustrated in Figure 2, aggregation is implemented through frame attention, composed of dual-stream DiT blocks Labs et al. (2025); Labs (2024), which aggregates intra-token information while fusing 3D positional prompts. Face token registration serves as the fundamental operation for fusing multiple inputs. In Figure 2, this is achieved via global attention, which aligns encoded face tokens to achieve 3D spatial registration and fusion. To enhance quality and accelerate convergence, we initialize our frame attention using weights from LAM's blocks. Global attention and frame attention are interleaved in a cascaded architecture, with a total of $L$ pairs employed to process face tokens, ultimately yielding tokens suitable for generating the 3DGS representation $\{\hat{h}_0, \cdots, \hat{h}_N\}$.

**Canonical 3DGS Model Fusion** Following the aggregation and registration through global attention and frame attention, we obtain processed tokens $\hat{h}_i$ corresponding to each frame. These features are then fed into a GS Head (i.e., a two-layer MLP with shared weights across tokens) to predict the target 3DGS attributes $g_i$. The point cloud $g_i^e$ derived by driving $g_i$ through Linear Blend Skinning (LBS) expression deformation is then rendered via Gaussian splatting to obtain the reconstructed face $\hat{I}_i$. Our approach extends beyond $g_i$; we further aggregate all $g_0, \cdots, g_N$:

$$g_f = \mathcal{U}(g_1, g_2, \cdots, g_N). \tag{3}$$

The fused $g_f$ integrates unique information from all perspectives (e.g., multi-view observations, diverse expressions), achieving optimal reconstruction quality. However, naive fusion would cause point cloud misalignment, ghosting artifacts, and color discrepancies. To address this, we introduce Landmark Tracking Loss and Sliced Fusion Loss to explicitly encourage proper alignment of Gaussian point clouds during aggregation and registration stages.

**3DGS Pruning** Although the 3DGS method achieves high-quality and real-time rendering, it often suffers from redundant memory consumption due to its explicit structure and tends to be more prone to overfitting because of the lack of smoothness bias in the neural network. This is especially problematic in our incremental reconstruction scenarios, where the number of GS points increases linearly with the number of input frames, leading to inefficient resource usage and limiting rendering speed. To address this, inspired by LP-3DGS Zhang et al. (2024b) and MaskGaussian Liu et al. (2025) we apply Gumbel-Softmax Jang et al. (2017) to sample one differentiable category, denoted as $\mathcal{M}_i \in \{0, 1\}$. Then we integrate masks directly within the rasterization framework, effectively decoupling Gaussian presence from attributes such as opacity and shape. This mechanism prunes over 50% of the GS primitives without degrading reconstruction quality, further improving rendering efficiency. To prune redundant 3D Gaussian primitives, we apply an L1 regularization term to the trainable mask, encouraging it to become sparse, formulated as $\mathcal{L}_{mask} = \frac{1}{N} \sum_{i=1}^{N} |m|$.

### 3.3 TRAINING STRATEGY

**Sliced Fusion Loss** To enable the model to take advantage of the richer information provided by multiple inputs, we introduce *Sliced Fusion Loss*, allowing $\mathcal{G}$ to handle arbitrary numbers of input frames. Specifically, during training, we randomly sample one frame from the input to obtain a single frame-wise Gaussian representation $g_i$. In parallel, we randomly select $N_{\text{sliced}}$ frames from the input, where $N_{\text{sliced}}$ is less than the total number of input frames for memory efficiency, and fuse them to construct a multi-frame Gaussian representation $g_{\text{sliced}}$. Both $g_i$ and $g_{\text{sliced}}$ are rendered into RGB images using the camera parameters, expression coefficients, and head poses of all input and target frames, and the corresponding losses are computed.

$$\hat{I}_i = \Psi\left(g_i, \pi_i, z_i^{\text{exp}}, z_i^{\text{pose}}\right), \tag{4}$$

$$\hat{I}_{\text{sliced}} = \Psi\left(g_{\text{sliced}}, \pi_i, z_i^{\text{exp}}, z_i^{\text{pose}}\right). \tag{5}$$

The overall loss function consists of two components: one supervises the rendering quality of the constructed 3D Gaussian head, and the other supervises the combination of frame-wise Gaussian representations to ensure consistency across frames.

| Method | 1 View | | | | | |
| --- | --- | --- | --- | --- | --- | --- |
| | PSNR ↑ | SSIM ↑ | LPIPS ↓ | Identity ↓ | FPS ↑ | Modeling Time (s) ↓ |
| LAM | 17.30 | 0.773 | 0.149 | 0.135 | 125 | **0.31** |
| MonoGaussianAvata | 11.83 | 0.631 | 0.620 | 0.432 | <10 | >100 |
| GaussianAvatars | 16.35 | 0.740 | 0.332 | 0.299 | <10 | >100 |
| FastAvatar | **20.08** | **0.860** | **0.143** | **0.116** | **240.17** | 1.33 |
| **Method** | 4 Views | | | | | |
| | PSNR ↑ | SSIM ↑ | LPIPS ↓ | Identity ↓ | FPS ↑ | Modeling Time (s) ↓ |
| LAM* | 16.69 | 0.743 | 0.204 | 0.167 | 45 | **0.39** |
| MonoGaussianAvatar | 12.71 | 0.798 | 0.437 | 0.368 | <10 | >100 |
| GaussianAvatars | 17.52 | 0.802 | 0.340 | 0.278 | <10 | >100 |
| FastAvatar | **22.12** | **0.880** | **0.094** | **0.098** | **101.62** | 4.22 |
| **Method** | 8 Views | | | | | |
| | PSNR ↑ | SSIM ↑ | LPIPS ↓ | Identity ↓ | FPS ↑ | Modeling Time (s) ↓ |
| LAM* | 16.59 | 0.718 | 0.235 | 0.206 | 24 | **0.43** |
| MonoGaussianAvatar | 13.11 | 0.650 | 0.493 | 0.298 | <10 | >100 |
| GaussianAvatars | 20.35 | 0.820 | 0.320 | 0.252 | <10 | >100 |
| FastAvatar | **22.19** | **0.880** | **0.093** | **0.097** | **52.28** | 8.56 |
| **Method** | 16 Views | | | | | |
| | PSNR ↑ | SSIM ↑ | LPIPS ↓ | Identity ↓ | FPS ↑ | Modeling Time (s) ↓ |
| LAM* | 16.49 | 0.697 | 0.265 | 0.238 | 13 | **0.69** |
| MonoGaussianAvatar | 15.81 | 0.721 | 0.406 | 0.202 | <10 | >100 |
| GaussianAvatars | 21.48 | 0.873 | 0.281 | 0.185 | <10 | >100 |
| FastAvatar | **22.29** | **0.881** | **0.092** | **0.095** | **17.65** | 26.06 |

Table 1: The quantitative comparison among FastAvatar, LAM He et al. (2025), MonoGaussianA-vatar Chen et al. (2024c), and GaussianAvatars Qian et al. (2024a) includes 3 critical metrics: Reconstruction quality (PSNR, SSIM, LPIPS); Modeling time: Duration required to reconstruct the 3DGS model; Inference speed: Animation rendering FPS of the output 3DGS model.

**Pixel Losses** The rendered RGB images are supervised using photometric losses against the corresponding ground truth target images:

$$\mathcal{L}_{rgb} = \left\| \hat{I}_i, I^{gt} \right\|_1 + \left\| \hat{I}_{\text{sliced}}, I^{gt} \right\|_1, \tag{6}$$

$$\mathcal{L}_{ssim} = \text{SSIM}(\hat{I}_i, I^{gt}) + \text{SSIM}(\hat{I}_{\text{sliced}}, I^{gt}), \tag{7}$$

We also incorporate perceptual losses to encourage the emergence of more high-frequency details:

$$\mathcal{L}_{lpips} = \text{LPIPS}(\hat{I}_i, I^{gt}) + \text{LPIPS}(\hat{I}_{\text{sliced}}, I^{gt}). \tag{8}$$

**Landmark Tracking Loss** Unlike novel view synthesis, canonical space registration is supervised directly on the input frames. The landmark tracking loss is introduced to encourage precise localization of facial landmarks throughout the input images:

$$\mathcal{L}_{track} = \sum_{j=1}^{M} \sum_{i=1}^{N} \| y_{j,i} - \hat{y}_{j,i} \|. \tag{9}$$

Our total loss is defined as follows:

$$\mathcal{L} = \lambda_1 \mathcal{L}_{rgb} + \lambda_2 \mathcal{L}_{ssim} + \lambda_3 \mathcal{L}_{lpips} + \lambda_4 \mathcal{L}_{track} + \lambda_5 \mathcal{L}_{mask}, \tag{10}$$

with $\lambda_1 = 0.8$, $\lambda_2 = 0.1$, $\lambda_3 = 0.1$, $\lambda_4 = 0.1$ and $\lambda_5 = 0.0005$.

## 4 EXPERIMENTS

### 4.1 TRAINING

We train FastAvatar on a multi-task dataset derived from NeRSemble Kirschstein et al. (2023), which contains multi-person, multi-camera videos with a wide range of facial expressions. To encourage

adaptability to diverse input settings, the dataset includes both monocular and multi-view subsets. Input-output pairs are constructed by sampling 16 frames each, either from a single video or from 12 camera views of the same subject. For each pair, a random subset of 1 to 16 input frames is further selected, enabling the model to robustly handle scenarios with sparse or varying numbers of input frames—such as real-world recordings with incomplete or irregular camera captures.

## 4.2 EXPERIMENTAL SETUPS

**Task.** We evaluate the model's ability to generate a 3D head avatar for unseen subjects from various types of input, including a single image, an unordered and arbitrary number of monocular video frames, and multi-view frames.

**Metrics.** We employ three paired-image metrics to measure the quality of individual rendered images: Peak Signal-to-Noise (PSNR), Structural Similarity Index (SSIM), and Learned Perceptual Image Patch Similarity (LPIPS). We also evaluate identity preservation by computing similarity using ArcFace Deng et al. (2019); Chen et al. (2020; 2024a) features.

**Baselines.** We compare FastAvatar with recent state-of-the-art systems for 3D head avatar generation across various tasks, including reconstruction from a single image, monocular video frames and multi-view frames. LAM He et al. (2025) A model that generates one-shot animatable Gaussian heads using a canonical Transformer with point-cloud representation and multi-scale cross-attention, enabling real-time, expression-consistent avatar animation and editing from a single image. Avat3r Kirschstein et al. (2025) A system for regressing animatable 3D head avatars from limited multi-view images by combining large reconstruction and foundation models with cross-attention layers to effectively model 3D facial dynamics and generalize across diverse data. Mono-GaussianAvatar Chen et al. (2024c) and GaussianAvatar Qian et al. (2024a) are two representative optimization-based 3D Avatar methods, both of which use FLAME as a proxy 3D model, similar to ours.

We conduct all comparison experiments on the same 48GB Nvidia RTX 4090 GPU. To ensure a fair comparison, we slightly modify the LAM renderer for single-shot input, enabling it to fuse information from multiple frames like FastAvatar, while keeping all other components consistent with the official repository. We retain the original model weights and confirm that its performance faithfully reflects the official version. We refer to this variant as LAM* in the following. Notably, with only 1 input frame, LAM* automatically reverts to the official LAM, which is designed for single-frame inputs, ensuring fair comparison.

## 4.3 QUALITATIVE COMPARISON

Comparative results against LAM, MonoGaussian, and GaussianAvatar are presented in Figure 3. We evaluate 4 distinct input configurations (1, 4, 8, and 16 views) by progressively increasing the number of input frames. Key observations indicate that while LAM yields performance roughly on par under single-view conditions, it fails to benefit from additional input views due to the lack of registration. Conversely, optimization-based methods exhibit significant performance degradation with sparse inputs (e.g., 1 view), though their reconstruction quality improves progressively as more views become available. FastAvatar consistently outperforms the baseline across all view settings (1∼16 views), while further enhancing its ability to capture fine-grained details—such as teeth gaps, wrinkles, and acne—as the number of views increases.

## 4.4 QUANTITATIVE COMPARISON

Table 1 presents a quantitative comparison of the four methods. All approaches demonstrate substantial improvements in reconstruction metrics with increasing input frames, with both MonoGaussianAvatar and GaussianAvatar exhibiting gains in both subjective assessments (i.e., LPIPS) and objective metrics (i.e., PSNR, SSIM). This reaffirms the critical importance of richer input data for high-fidelity reconstruction. However, LAM shows an inverse relationship: as its input views increase, quantitative performance degrades, which can also be illustrated in Figure 4. Although LAM achieves impressive visual quality with single image (LPIPS: 0.149), its generative bias introduces pose and expression artifacts that compromise objective measurements. FastAvatar achieves highly

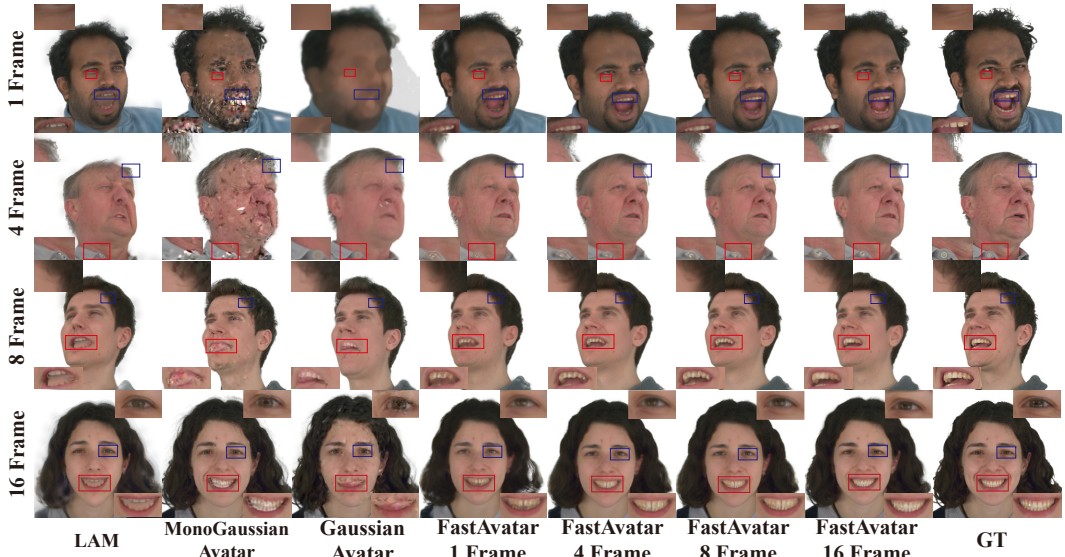

Figure 3: We benchmark FastAvatar against representative optimization-based methods (Mono-Gaussian Avatar Chen et al. (2024c), GaussianAvatar Qian et al. (2024a)) and feedforward approaches (LAM He et al. (2025)). Our results demonstrate the performance evolution across methods as the number of input views (referring to input images number) increases. Please zoom in for a better view.

competitive growth across both subjective and objective dimensions, validating our core hypothesis. Through architectural and training innovations, FastAvatar establishes an optimal equilibrium between generative capability (hallucinating plausible details under sparse inputs) and reconstruction fidelity (strict adherence to observed data given sufficient views).

## 4.5 INCREMENTAL RECONSTRUCTION

FastAvatar enables incremental improvement by accepting input sequences of any length and order. As more observations are added, the reconstruction quality continues to improve. In contrast, existing methods often require a fixed number of input frames, which reduces flexibility and may result in data loss. FastAvatar's incremental design thus ensures both versatility and efficient data usage.

As illustrated in Figure 4, our method achieves superior expressiveness and rendering fidelity in avatar generation compared to the baselines, and demonstrates robust performance even for subjects wearing accessories.

Moreover, increasing the number of input views further improves the reconstruction of fine-grained details, such as hair and teeth gaps. This incremental reconstruction allows the model to overcome the limitations of restricted viewpoints by leveraging more informative input frames—a capability that is difficult to achieve for models with fixed input forms. For example, in Figure 4, the character's left-ear earring is not sufficiently observed with a small number of input views, but it is reliably reconstructed as the number of views increases.

Unlike the sparse and randomly sampled data used in our main experiments, real-world sequences are highly continuous. Processing all frames with Global Attention imposes prohibitive computational and memory costs. A naive solution is to uniformly sample 16 frames, but this risks missing important information present in the remaining frames. Inspired by FramePack Zhang et al. (2025), we retain the 16 uniformly sampled frames as sparse inputs and compress all remaining frames into an aggregated token representation, which is then treated as an additional input. This preserves complementary details while keeping computation tractable, enabling FastAvatar to process hundreds of frames in a single feed-forward pass. Figure 5 shows that adding the compressed frames improves fine-grained details.

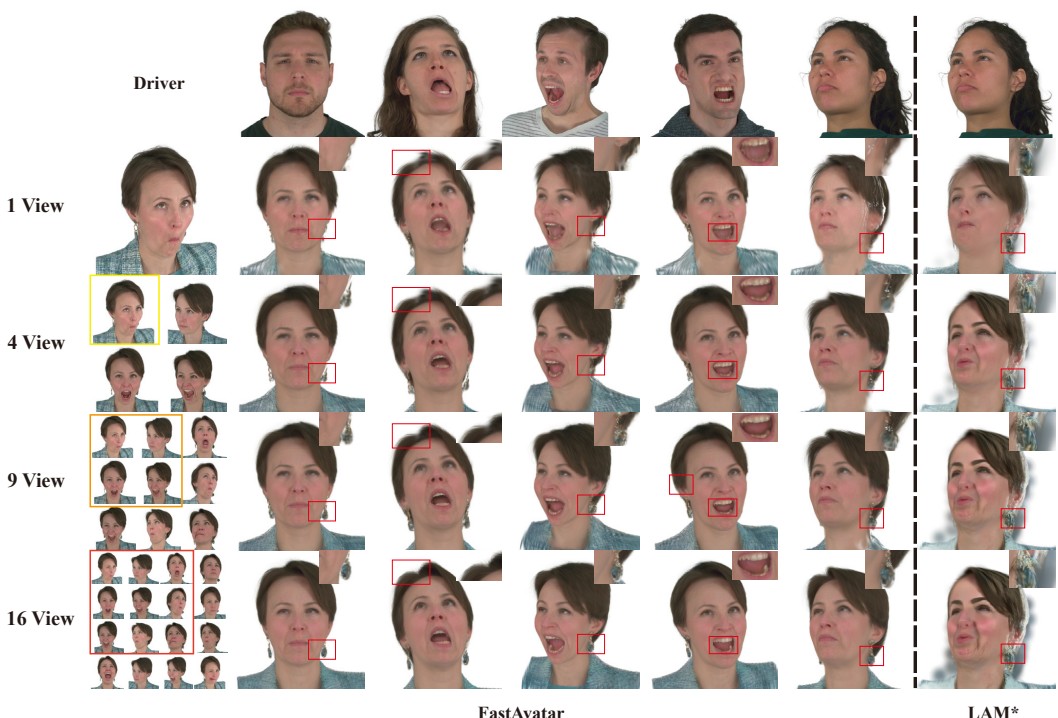

Figure 4: Reconstruction quality as the number of input observations increases. More observations improve reconstruction quality.

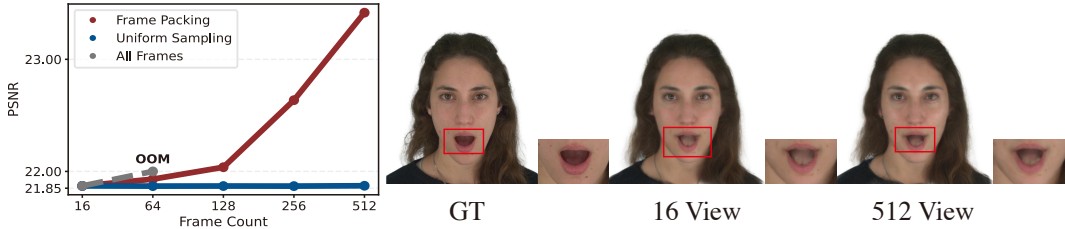

Figure 5: Performance on longer input sequences. Starting from strong reconstructions using only the 16 sparse input frames, incorporating the compressed additional frames further enhances fine-grained details (e.g., the oral cavity, which is absent in most frames). While uniform sampling fails to achieve this improvement, feeding all frames leads to OOM.

### 4.6 MULTI-VIEW OBSERVATIONS RECONSTRUCTION

To further evaluate FastAvatar's performance on the Multi-view Observations Reconstruction task, we create multi-view few-shot 3D head avatars for subjects from the Ava256 Martinez et al. (2024) dataset, which was not used during training. We compare our results with those of the state-of-the-art method Avat3r Kirschstein et al. (2025) and use the results provided in its original paper to ensure a fair comparison (since its implementation is not publicly available, we ensure fairness by directly using results from the Avat3r paper). The qualitative results are shown in Figure 6. Note that we only used images from Ava256 for tracking and to obtain FLAME parameters, without utilizing the provided informed encodings. Nevertheless, FastAvatar still achieves highly competitive results and benefits from additional multi-view inputs, producing more detailed reconstructions. Close inspection reveals that Avat3r fails to preserve facial identity accurately, reconstructing consistently wider facial structures than observed in source inputs.

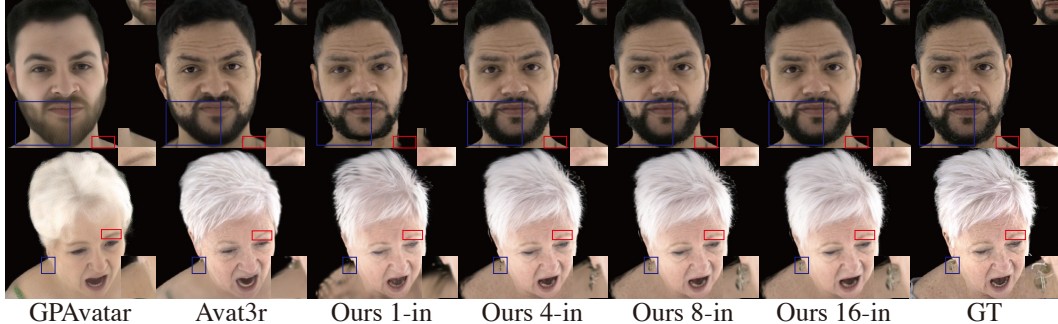

| GPAvatar | Avat3r | Ours 1-in | Ours 4-in | Ours 8-in | Ours 16-in | GT |

Figure 6: Visual comparison with Avat3r and GPAvatar Chu et al. (2024). Please zoom in for a clearer view.

| Method | 1 View | | | | | | 4 View | | | | | |
| | L1↓ | PSNR↑ | SSIM↑ | LPIPS↓ | Identity↓ | # GS (K)↓ | L1↓ | PSNR↑ | SSIM↑ | LPIPS↓ | Identity↓ | # GS (K)↓ |
| --- | --- | --- | --- | --- | --- | --- | --- | --- | --- | --- | --- | --- |
| w/o sliced fusion loss | 0.0373 | 20.47 | 0.861 | **0.123** | 0.158 | 20.7 | 0.0345 | 21.69 | 0.857 | 0.131 | 0.138 | 82.8 |
| w/o tracking loss | **0.0372** | **20.93** | **0.863** | 0.140 | 0.172 | 13.0 | 0.0322 | 21.64 | 0.866 | 0.120 | 0.128 | 41.2 |
| w/o global attention | 0.0922 | 15.40 | 0.760 | 0.238 | 0.400 | **10.3** | 0.0462 | 19.49 | 0.828 | 0.162 | 0.270 | 40.1 |
| w/o GS fusion | 0.0379 | 20.31 | 0.857 | 0.136 | **0.138** | 12.8 | 0.0467 | 18.94 | 0.838 | 0.157 | 0.185 | **12.5** |
| w/o GS pruning | 0.0380 | 20.32 | 0.857 | 0.137 | 0.144 | 21.7 | 0.0322 | 21.67 | 0.868 | 0.112 | 0.130 | 86.7 |
| Ours full | 0.0379 | 20.31 | 0.857 | 0.136 | 0.148 | 12.8 | **0.0303** | 21.86 | 0.871 | 0.107 | 0.118 | 42.8 |
| **Method** | 8 View | | | | | | 16 View | | | | | |
| | L1↓ | PSNR↑ | SSIM↑ | LPIPS↓ | Identity↓ | # GS (K)↓ | L1↓ | PSNR↑ | SSIM↑ | LPIPS↓ | Identity↓ | # GS (K)↓ |
| w/o sliced fusion loss | 0.0386 | 21.12 | 0.849 | 0.144 | 0.151 | 165.4 | 0.0417 | 20.62 | 0.839 | 0.159 | 0.180 | 330.5 |
| w/o tracking loss | 0.0320 | 21.61 | 0.867 | 0.119 | 0.124 | 78.6 | 0.0322 | 21.66 | 0.865 | 0.123 | 0.129 | 164.2 |
| w/o global attention | 0.0413 | 19.97 | 0.835 | 0.156 | 0.223 | 78.0 | 0.0405 | 20.06 | 0.830 | 0.167 | 0.210 | 155.7 |
| w/o GS fusion | 0.0574 | 17.44 | 0.823 | 0.179 | 0.227 | **12.5** | 0.0682 | 16.25 | 0.811 | 0.196 | 0.259 | **12.4** |
| w/o GS pruning | 0.0326 | 21.63 | 0.868 | 0.110 | 0.130 | 173.4 | 0.0327 | 21.61 | 0.867 | 0.110 | 0.136 | 346.8 |
| Ours full | **0.0297** | **21.95** | **0.871** | **0.103** | **0.118** | 77.0 | **0.0293** | **22.04** | **0.876** | **0.102** | **0.118** | 138.9 |

Table 2: Ablation studies on key components of FastAvatar. The appendix visualizations are strongly recommended for better understanding.

## 4.7 ABLATION STUDY

To validate the effectiveness of each component in our method, we conduct both quantitative and qualitative ablation studies. The qualitative visualizations are provided in the appendix for space considerations. As shown in Table 2, Global Attention is crucial for coordinating inter-frame information, while GS Fusion aggregates the GS points from each frame into a unified representation. Sliced Fusion Loss and Tracking Loss supervise GS registration, enforcing structural consistency and temporal coherence. Without these components, newly introduced frames fail to provide reliable information, leading to degraded registration and blurred outputs as the number of frames increases. Meanwhile, Gaussian Pruning removes redundant primitives, slightly improving performance while substantially accelerating rendering. Together, these mechanisms ensure effective inter-frame coordination, accurate registration, and efficient rendering.

## 5 CONCLUSION

In this paper, we present FastAvatar, a feed-forward 3D avatar reconstruction framework capable of constructing a high-quality animatable 3DGS avatar within seconds. Distinct from existing approaches, FastAvatar demonstrates a unique capability for incremental avatar reconstruction – flexibly leveraging incoming observations to progressively enhance reconstruction quality. We contend this represents a promising research trajectory. Three pivotal innovations enable this functionality: Alternating Attention, augmented with fine-grained expression and pose encodings, achieves high-precision registration of unordered data; The proposed Landmark Tracking Loss and Sliced Fusion Loss facilitate robust fusion of multiple 3DGS representations for superior modeling fidelity. Experimental validation confirms FastAvatar's potential in these dimensions.

## 6 ACKNOWLEDGMENTS

Kairui Feng was supported by the National Natural Science Foundation of China (Grant No. 62088101), Shanghai Municipal Science and Technology Major Project (Grant No. 2021SHZDZX0100), Explorer Program (Grant No. 24TS1401600), and Xiaomi Foundation.

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

## A    LLM Usage

Large Language Models (LLMs) were used solely to assist in refining the manuscript's language, improving readability, and ensuring clarity, including sentence rephrasing and grammar checking. The LLM did not contribute to the research ideas, methodology, experiments, or data analysis. The authors take full responsibility for the scientific content and confirm that all LLM-assisted text adheres to ethical guidelines, with no contribution to plagiarism or misconduct.

## B    Ethics Statement

The proposed FastAvatar framework follows the same data assumptions and usage boundaries as existing 3D avatar reconstruction and neural rendering methods. It does not introduce new mechanisms that lower the barrier to unauthorized identity reconstruction, nor does it relax requirements on input data quality. In practice, the method still relies on clean and identity-consistent multi-frame input, which inherently limits large-scale or covert misuse.

All experiments are conducted on publicly available datasets with appropriate licenses, and no private or sensitive data are collected. The intended applications of FastAvatar lie in areas such as AR/VR telepresence, digital content creation, and human–computer interaction, where user consent is typically explicit. We emphasize that responsible deployment requires ensuring that the method is not applied to reconstruct individuals without consent or to generate deceptive or impersonating content.

## C    Reproducibility

In this section, we provide more implementation details of FastAvatar, including data preparation and model architecture. Furthermore, our code will be released after the paper is accepted.

### C.1    Data Preparation

Our training utilizes the Nersemble dataset. Initially, FLAME tracking is applied to extract FLAME parameters and camera poses, which serve as inputs for subsequent training stages. From the original Nersemble data, we extract 521 distinct video clips, and sample the frames at 15 FPS. Cameras with poor face tracking quality were excluded, remaining 12 cameras. The processed data was sampled twice to construct the final dataset: first, sampling frames within the same video sequence, and second, performing random sampling across all shots of the same action sequence. These two sampling strategies collectively support training the unified task. To enhance the stability of training and testing, we randomly assign the processed figures' backgrounds to black, white, or gray. Note that, to enhance the model's generative capability, all expression parameters, poses, and related data used during inference differ from the input data.

| | Hyperparameter | Value |
|---|---|---|
| Encoder | DINOv2 patch size | $14 \times 14$ |
| | #expression token MLP layers | 2 |
| | #camera token MLP layers | 2 |
| | Expression Token MLP activation | GELU |
| | Camera Token MLP activation | GELU |
| | Output dimension | 1024 |
| | Input image resolution | $504 \times 504$ |
| AA | #Frame Attn Layers | 10 |
| | #Global Attn Layers | 10 |
| | Hidden dimension | 1024 |
| | Order | [Global, Frame] |

Table 3: Hyperparameters. Where AA represents Alternation Attention

| Noise | L1↓ | PSNR↑ | SSIM↑ | LPIPS↓ | Identity↓ |
|-------|------|-------|-------|--------|-----------|
| 1 px | 0.0264 | **22.50** | **0.873** | **0.096** | 0.100 |
| 4 px | 0.0268 | 22.38 | 0.872 | 0.098 | **0.099** |
| 8 px | 0.0273 | 22.22 | 0.870 | 0.098 | 0.103 |
| 16 px | 0.0277 | 22.10 | 0.869 | 0.099 | 0.102 |
| 32 px | 0.0280 | 22.02 | 0.869 | 0.099 | 0.105 |
| Ours | **0.0263** | **22.50** | 0.872 | **0.096** | **0.099** |

Table 4: Ablation studies on FLAME tracking. We evaluate the robustness of FastAvatar under varying levels of landmark perturbation during FLAME tracking.

The accuracy of FLAME tracking primarily depends on the precision of detected facial landmarks, as the FLAME parameters are typically estimated by optimizing the model to fit these landmarks. However, the reliability of such proxy models (e.g., FLAME and other 3DMMs) is inherently constrained by factors such as limited representational capacity and sensitivity to landmark quality. To assess how these factors affect our method, we include an additional ablation experiment that injects controlled noise into the facial landmarks. The results (Table 4) demonstrate that FastAvatar remains robust under reasonable perturbations, indicating that strict accuracy in FLAME tracking is not required.

## C.2 TRAINING

In table 3, we present the most important hyperparameters for training FastAvatar. We train the model by optimizing the training loss with the AdamW optimizer for 150K iterations. We use a cosine learning rate scheduler with learning rate of 4e-5. The training runs on 8 H100 GPUs over 14 days. The substantial accumulation of Gaussian points across multiple input frames leads to high GPU memory consumption during training. To address this, we adopt bfloat16 precision and gradient checkpointing for improved memory and computational efficiency.

## C.3 FRAMEPACK

While 16 sparse frames suffice for high-quality 3D head reconstruction, quality degrades when viewpoint coverage is incomplete—e.g., mouth-opening expressions fail without intra-oral views. However, full global attention over all frames is prohibitively expensive.

We address this with a two-tier token design inspired by FramePack. We designate 16 base frames whose DINO features are kept at full spatial resolution, and compress all remaining frames with a learned $k \times k$ strided convolution ($k=8$), reducing each to $1/k^2$ of its original token count. Within each alternating-attention layer, frame attention cross-attends each frame's point cloud with its own image tokens independently—base frames at full resolution, compressed frames through a separate set of weights at reduced resolution. Global attention then concatenates all tokens across both tiers and applies RoPE-based self-attention jointly. This provides coverage from arbitrarily many extra viewpoints at a cost sublinear in total frame count, enabling incremental reconstruction from hundreds of input frames.

## D MORE RESULTS

In this section, we present additional results of FastAvatar, including its performance on a broader set of video sequences and its generalization to real-world daily-captured data.

**More Comparison** We provide additional qualitative results comparing FastAvatar and baseline models in both self-reenactment and cross-reenactment settings. As shown in Figure 15, Figure 16, and Figure 14. FastAvatar outperforms the baselines. Optimization-based 3D avatar methods fail to achieve satisfactory results with sparse inputs, while LAM often exhibits unrealistic details and significant pose inaccuracies. The advantage of FastAvatar becomes even more evident in the cross-reenactment setting, where the subject's identity and camera pose exhibit large discrepancies. We further evaluate FastAvatar against additional competitive methods. The results are presented in Figure 7 and Figure 8.

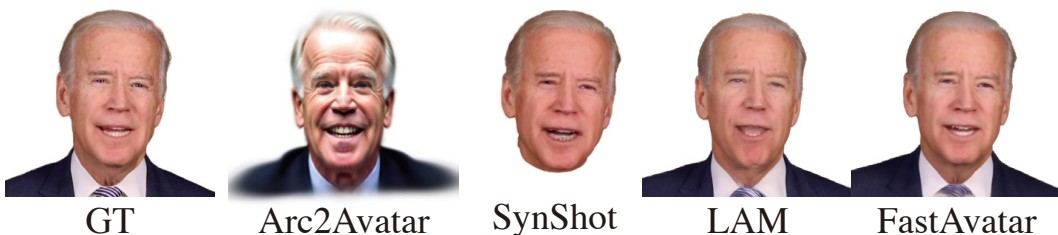

Figure 7: Qualitative results on the INSTA dataset. LPIPS scores: Ours (**0.1332**), LAM (0.1479), Arc2Avatar (0.4585), SynShot (0.1523). Identity scores: Ours (**0.076**), LAM (0.124), Arc2Avatar (0.411), SynShot (0.115).

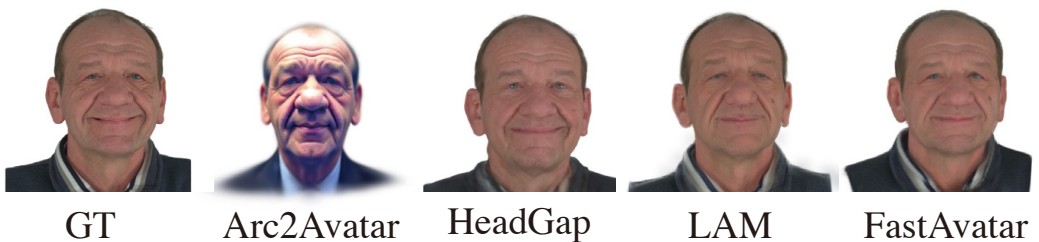

Figure 8: Qualitative results on the Nersemble dataset. LPIPS scores: Ours (**0.1267**), LAM (0.1608), Arc2Avatar (0.4665), HeadGap (0.1592). Identity scores: Ours (**0.070**), LAM (0.097), Arc2Avatar (0.308), HeadGap (0.101).

**Generalization to Wide-Range Viewpoints**   The Nersemble training set contains only 12 constrained camera views. To evaluate the robustness of our method, we test it on a much wider range of viewpoints. As shown in Figure 12, the model maintains high-fidelity reconstruction across all novel views, demonstrating strong wide-range generalization. For comparison, we include the results of LAM in Figure 13. The results demonstrate that FastAvatar outperforms the state-of-the-art across a wide range of viewpoints.

**Ablation Study**   We further highlight the role of the key components in incremental reconstruction. As illustrated in Figure 9, removing these components prevents fine details from being properly aligned, leading to noticeable artifacts and blurred regions. Although the landmark tracking loss only supervises 68 facial landmark points, it still provides strong guidance for Gaussian registration, effectively assisting the model in aligning new frames during incremental updates. Together with the Sliced Fusion Loss, it ensures that additional observations can be accurately fused, enabling consistent refinement of the reconstructed avatar. Global Attention enables the model to leverage inter-frame dependencies, integrating complementary features from multiple frames; without it, information remains localized, and cross-frame consistency cannot be achieved. GS Fusion consolidates per-frame Gaussians into a coherent representation, allowing the model to maintain consistency across frames. Finally, Gaussian Pruning removes redundant primitives, slightly improving performance while significantly accelerating rendering, enabling efficient incremental updates even for long sequences.

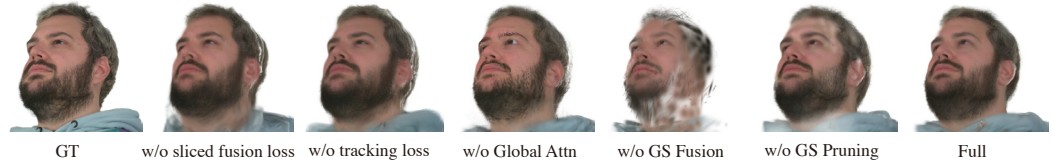

Figure 9: Comparison of visual effects of model reconstruction after removing the key components.

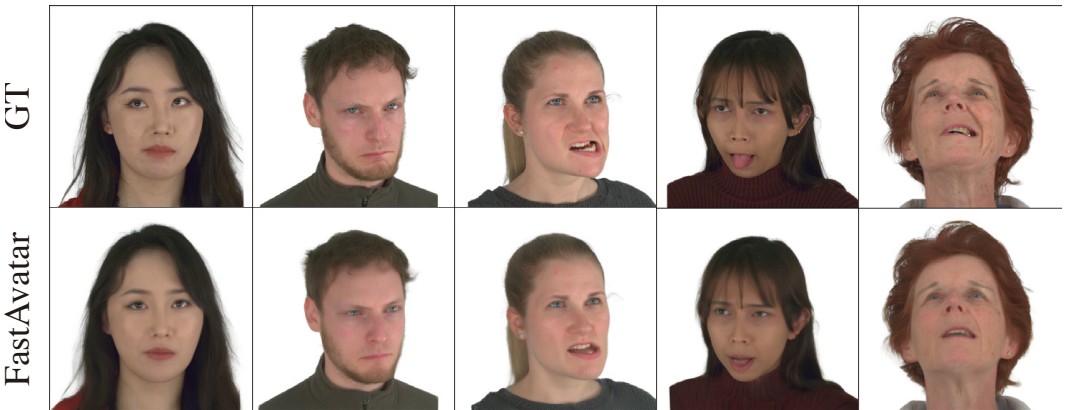

Figure 10: Typical failure cases: FastAvatar relying on LBS and FLAME-based encodings, struggles with complex facial muscle dynamics, fine-grained details (e.g., wrinkles), eye-gaze movements, and structures outside the FLAME topology such as the tongue.

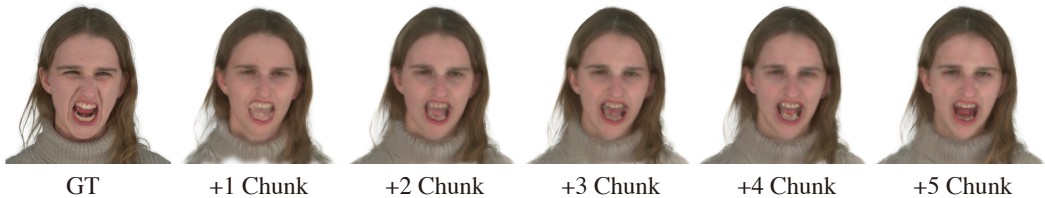

|  GT | +1 Chunk | +2 Chunk | +3 Chunk | +4 Chunk | +5 Chunk |

Figure 11: As the streaming input is progressively incorporated, the reconstruction of the oral cavity—largely invisible in most chunks—gradually improves while structural consistency is maintained in other regions, enabling incremental reconstruction.

**Streaming Incremental Reconstruction** FastAvatar is also capable of streaming incremental reconstruction, meaning the model continuously updates and refines the 3DGS representation as new video frames are received. To achieve this, we adopt a sliding-window approach, where each window contains 16 frames with a 4-frame overlap for registering incoming frames. Thanks to the Alternating Attention design, new frames can build upon the Gaussians predicted by the previous model to produce more informative reconstructions. Figure11 demonstrates this: starting from a single-view image, as additional views are provided (excluding the test view), reconstruction quality improves overall, including at previously unseen angles, thus realizing streaming incremental reconstruction.

**Limitations** First, our method relies on LBS and FLAME-based encodings to drive 3D head avatar motion, which limits the representation of complex facial muscle dynamics. As a result, the model has difficulty reproducing fine-grained muscle-dependent details such as wrinkles and also cannot accurately capture eye-gaze movements, often defaulting to an average direction. Furthermore, because the Gaussians are anchored to FLAME vertices, the model is unable to represent structures outside the FLAME topology, including the tongue. Figure 10 presents representative failure cases.

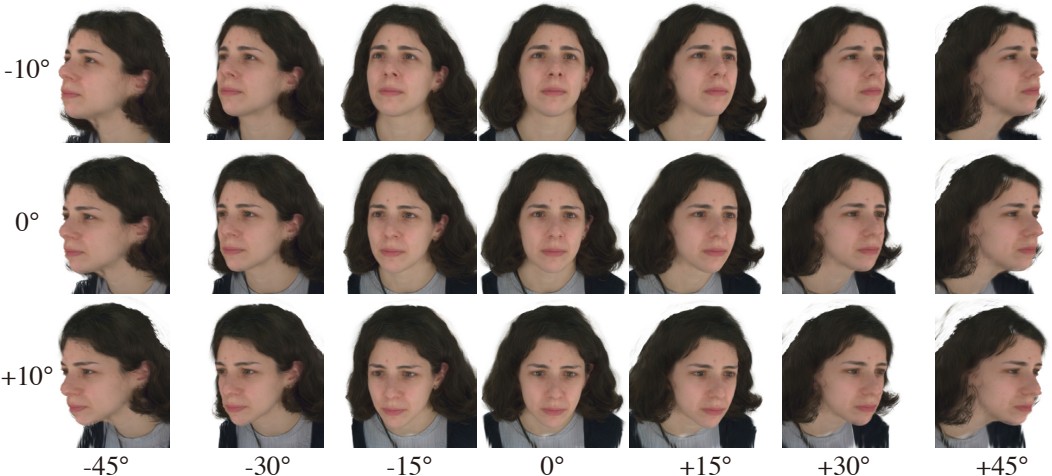

Figure 12: Generalization to wide-range viewpoints. FastAvatar achieves high-fidelity reconstruction across 14 novel viewpoints that are entirely outside the training set.

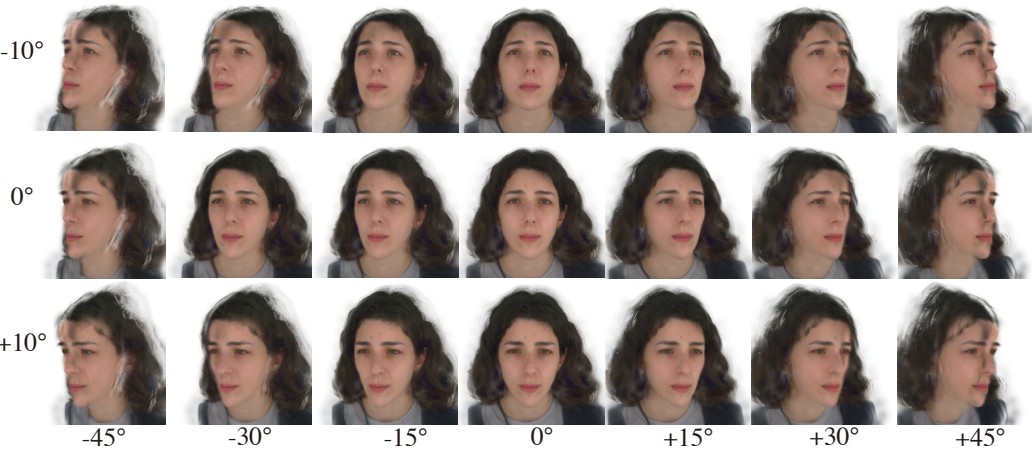

Figure 13: The performance of LAM on wide-range viewpoints.

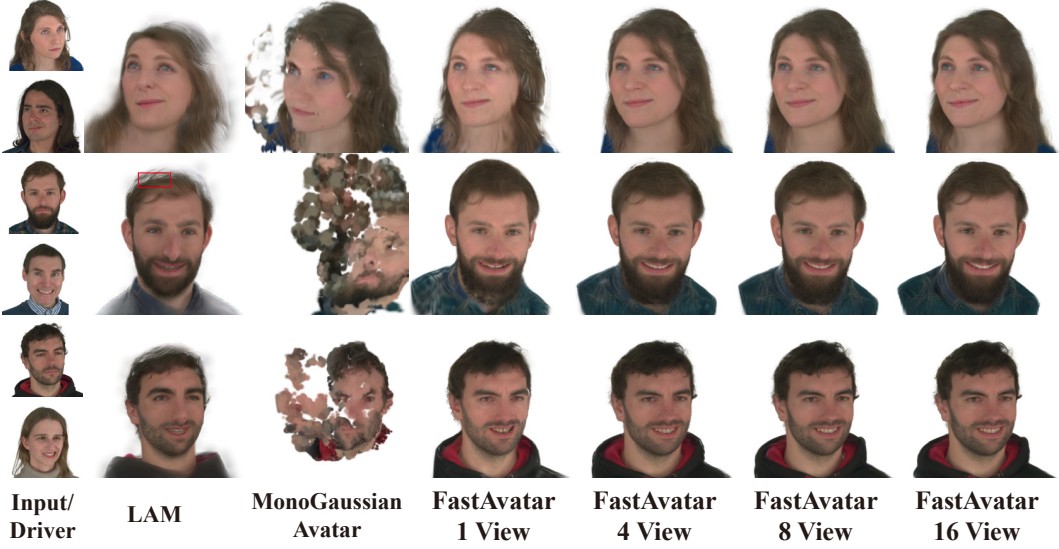

Figure 14: Additional Comparisons with Baseline Methods (cross-reenactment).

**GT**

**LAM**

**MonoGaussianAvatar**

**GaussianAvatars**

**FastAvatar**

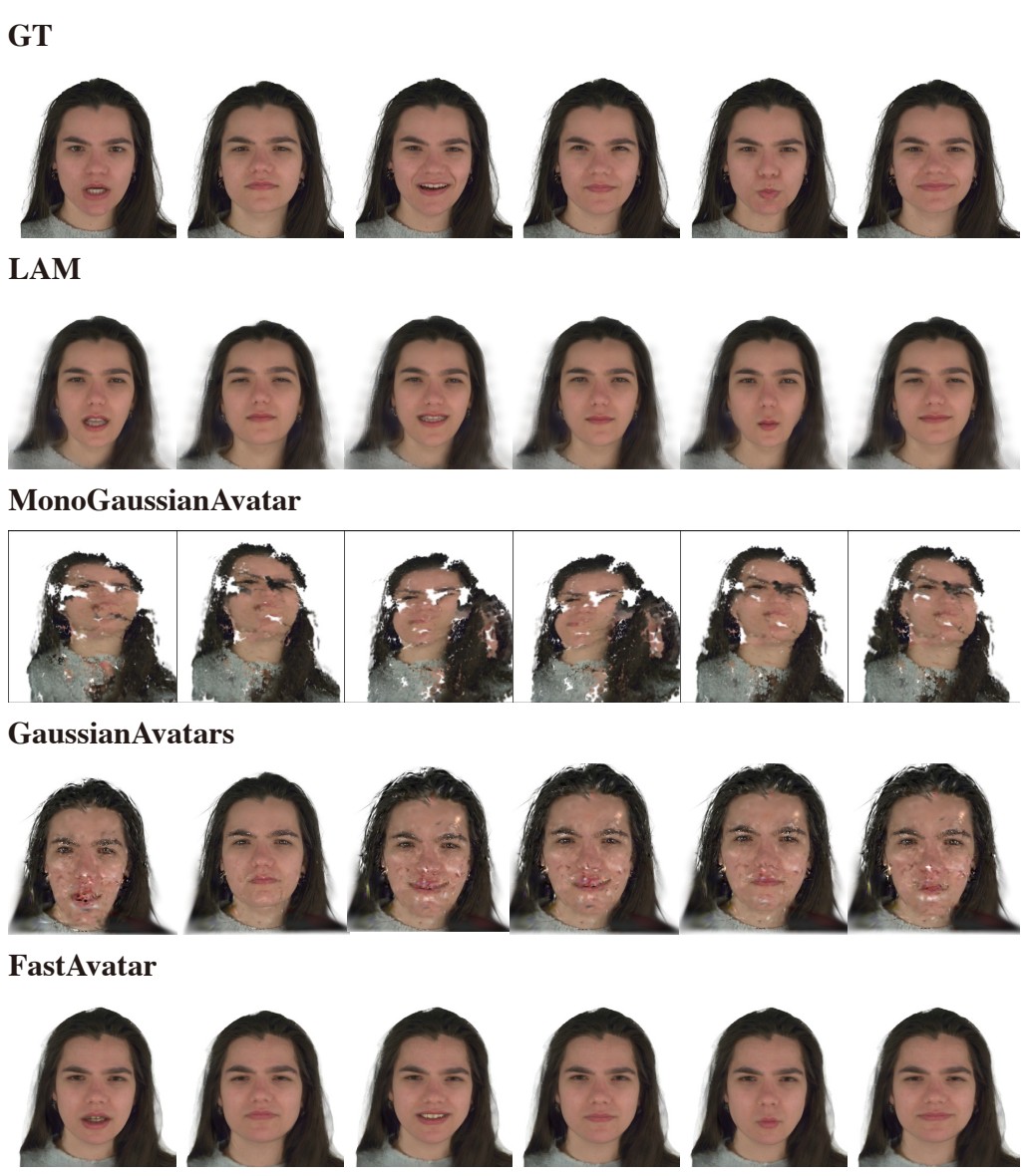

Figure 15: Additional Comparisons with Baseline Methods (self-reenactment part 1).

**GT**

**LAM**

**MonoGaussianAvatar**

**GaussianAvatars**

**FastAvatar**

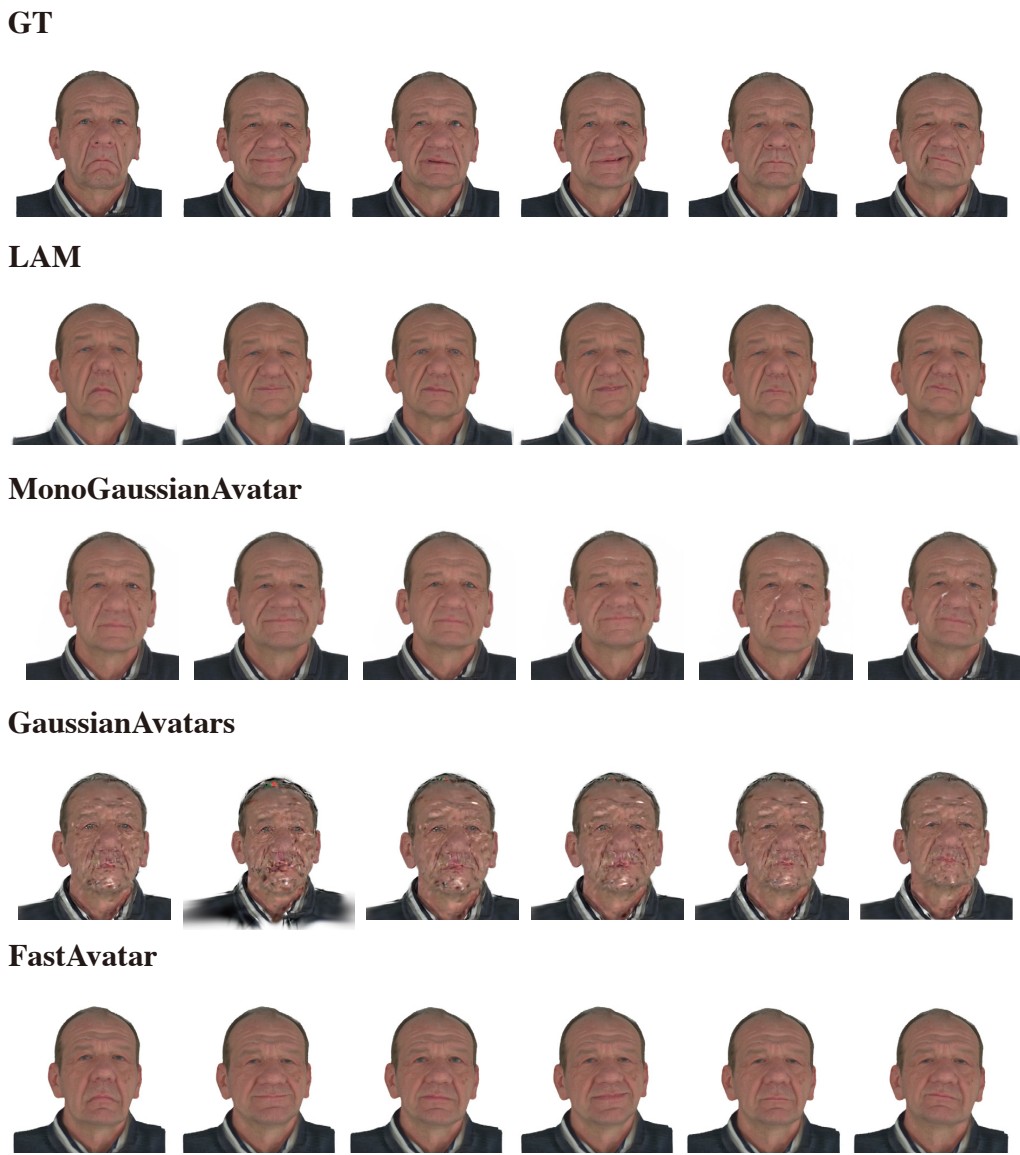

Figure 16: Additional Comparisons with Baseline Methods (self-reenactment part 2).

