# OpenReview forum: "FastAvatar: Towards Unified and Fast 3D Avatar Reconstruction with Large Gaussian Reconstruction Transformers"
_ICLR.cc/2026/Conference — ICLR 2026 Poster_

### Official Review · Reviewer_8Upx · 2025-10-24

**Soundness:** 3
**Presentation:** 2
**Contribution:** 2
**Rating:** 4
**Confidence:** 4

**Summary:**

This paper presents a feedforward framework for rapid 3D avatar reconstruction based on 3D Gaussian Splatting (3DGS). It can generate high-quality avatars from diverse inputs (single image, multi-view, or video) within seconds using a single unified model. The proposed Large Gaussian Reconstruction Transformer (LGRT) introduces (1) a 3DGS transformer for canonical reconstruction, (2) multi-granular guidance to handle pose and expression variations, and (3) incremental Gaussian aggregation for quality refinement. Experiments demonstrate that FastAvatar achieves good quality and speed compared to existing approaches.

**Strengths:**

1. Supports the reconstruction of high-quality 3D heads from various input types, including a single image, multiple frames, or multi-view observations.
2. Designs a full-attention mechanism and a differentiable sampling strategy to effectively fuse image information from multiple inputs.
3. Proposes a sliced fusion loss to further optimize reconstruction quality.

**Weaknesses:**

1. Ambiguous description: In Lines 200–209, the authors state that h_i represents the concatenated feature of a single image I_i with its expression z_i^{exp}. However, in Figure 2, h_i appears to denote the feature corresponding to different image/expression pairs under the same camera pose. The authors should clarify this inconsistency.
2.  Missing ablation studies: In Lines 234–236, the authors claim that the naïve fusion strategy underperforms compared to their proposed Gumbel-Softmax sampling. However, no ablation study is provided to substantiate this claim. The authors are encouraged to include experiments quantifying the contribution of the Gumbel-Softmax sampling to the final performance.
3. Lack of visual comparisons: The supplementary video only presents results from the proposed method. Including visual comparisons with other state-of-the-art approaches would more convincingly demonstrate the superiority of the proposed technique.
4. Unclear contribution of key modules: It remains unclear how much the full-attention mechanism and the final Canonical 3DGS Model Fusion contribute to multi-frame aggregation. The authors are encouraged to perform an ablation study or provide quantitative evidence demonstrating their effectiveness. Since multi-view and multi-frame fusion form a core contribution of this paper, a more detailed analysis and discussion of these components would significantly strengthen the work.
5. Lack of ethics statements.

**Questions:**

1. Why does your method achieve faster inference speed than LAM? Since multi-view fusion involves full attention and sampling operations, wouldn’t this introduce significant additional computational overhead?

**Details Of Ethics Concerns:**

The proposed FastAvatar framework enables the rapid reconstruction of realistic 3D human avatars from images or videos. While the technical contribution is significant, the methodology could be potentially misused to generate virtual representations of individuals without their consent or to create deepfake content, which may lead to privacy violations, identity misuse, or other social harms.

---

> ### Author Response · Authors · 2025-11-28
> **Response to Reviewer 8Upx**
>
> **Weakness 1**: There is no inconsistency between the figure and the text. In the figure, we use the same color to indicate camera tokens, but this does not mean that all camera tokens are identical. Each camera token is separately encoded from the camera parameters of the corresponding image. This can be observed from the leftmost input images in **Figure 2**, where each row corresponds to a different camera pose. The same principle applies to expression and pose tokens, which are also individually encoded for each frame.
>
> | Method | PSNR ↑ | SSIM ↑ | LPIPS ↓ | ID ↓ | #GS(K) ↓ | PSNR ↑ | SSIM ↑ | LPIPS ↓ | ID ↓ | #GS(K) ↓ |
> |--------|--------|--------|----------|-------|-----------|--------|--------|----------|-------|-----------|
> |        | **1 frame** | | | | | **4 frames** | | | | |
> | w/o sliced fusion loss | _20.47_ | _0.861_ | **0.123** | 0.158 | 20.7 | _21.69_ | 0.857 | 0.131 | 0.138 | 82.8 |
> | w/o tracking loss | **20.93** | **0.863** | 0.140 | 0.172 | 13.0 | _21.64_ | 0.866 | 0.120 | _0.128_ | 41.2 |
> | w/o global attention | 15.40 | 0.760 | 0.238 | 0.400 | **10.3** | 19.49 | 0.828 | 0.162 | 0.270 | 40.1 |
> | w/o GS fusion | 20.31 | 0.857 | _0.136_ | **0.138** | _12.8_ | 18.94 | 0.838 | 0.157 | 0.185 | **12.5** |
> | w/o GS pruning | 20.32 | 0.857 | 0.137 | _0.144_ | 21.7 | _21.67_ | _0.868_ | _0.112_ | 0.130 | 86.7 |
> | **Ours full** | 20.31 | 0.857 | _0.136_ | 0.148 | _12.8_ | **21.86** | **0.871** | **0.107** | **0.118** | _42.8_ |
>
> ---
>
> | Method | PSNR ↑ | SSIM ↑ | LPIPS ↓ | ID ↓ | #GS(K) ↓ | PSNR ↑ | SSIM ↑ | LPIPS ↓ | ID ↓ | #GS(K) ↓ |
> |--------|--------|--------|----------|-------|-----------|--------|--------|----------|-------|-----------|
> |        | **8 frames** | | | | | **16 frames** | | | | |
> | w/o sliced fusion loss | 21.12 | 0.849 | 0.144 | 0.151 | 165.4 | 20.62 | 0.839 | 0.159 | 0.180 | 330.5 |
> | w/o tracking loss | _21.61_ | 0.867 | 0.119 | _0.124_ | 78.6 | _21.66_ | 0.865 | 0.123 | _0.129_ | 164.2 |
> | w/o global attention | 19.97 | 0.835 | 0.156 | 0.223 | 78.0 | 20.06 | 0.830 | 0.167 | 0.210 | 155.7 |
> | w/o GS fusion | 17.44 | 0.823 | 0.179 | 0.227 | **12.5** | 16.25 | 0.811 | 0.196 | 0.259 | **12.4** |
> | w/o GS pruning | _21.63_ | _0.868_ | _0.110_ | 0.130 | 173.4 | 21.61 | 0.867 | 0.110 | 0.136 | 346.8 |
> | **Ours full** | **21.95** | **0.871** | **0.103** | **0.118** | _77.0_ | **22.04** | **0.876** | **0.102** | **0.118** | _138.9_ |
>
> ---
>
>
> **Weakness 2**: This was a typo. The Gumbel-sigmoid module is primarily used for GS pruning, where it computes a mask based on the importance scores of GS points to further accelerate rendering. We have added the corresponding ablation results in **Table 2** and **Figure 9**. As shown, GS pruning can remove a large number of GS points while preserving rendering quality. This significantly lowers the barrier to practical deployment, enabling most 3D avatar reconstruction scenarios to run efficiently even on a 48 GB RTX 4090 GPU.
>
> **Weakness 3**: In the supplementary video, we further include comparisons with LAM, a feed-forward state-of-the-art approach for single-image 3D Avatar reconstruction. The results demonstrate that FastAvatar also achieves superior visual quality on in-the-wild data.
>
> **Weakness 4**: We conducted a more comprehensive ablation study covering additional key components and varying numbers of input frames (see **Table 2** and **Figure 9**). When either the Global Attention or the GS points fusion module is removed, the model performance degrades as the number of input frames increases. This indicates that Global Attention is essential for aggregating multi-frame information; without cross-frame interaction, additional frames introduce more noise rather than useful cues.
> Moreover, the proposed GS points fusion is also critical. After Alternation Attention, the multi-frame features become well coordinated, with each frame tending to “focus on its own role” rather than letting the GS points of a single frame absorb information from all others. Without the fusion step, this coordinated structure collapses, leading to severe information loss and noticeably worse results.
>
> **Weakness 5**: We added the Ethics Statement section in the Appendix.
>
> **Question**: LAM is indeed faster than FastAvatar in terms of modeling time; however, its FPS measured based on rendering time is lower than that of FastAvatar. FastAvatar takes longer to model because it incorporates both Frame Attention and Global Attention. The difference arises during rendering: LAM’s official implementation renders frames one at a time (see https://github.com/aigc3d/LAM/blob/master/lam/models/modeling_lam.py line 314). Our pipeline renders all frames together—and in chunks when necessary—resulting in much higher FPS. This optimization pertains to engineering efficiency rather than algorithmic novelty, so we did not highlight it extensively. Moreover, GS pruning eliminates a significant number of redundant GS points (see **Table 2**), resulting in faster rendering.

---

### Official Review · Reviewer_Fo9q · 2025-10-27

**Soundness:** 3
**Presentation:** 2
**Contribution:** 3
**Rating:** 4
**Confidence:** 4

**Summary:**

The paper presents a feed-forward 3D avatar head reconstruction framework that performs high-quality 3DGS-based modeling within seconds from arbitrary-length inputs (single/multi-view or video). They propose Large Gaussian Reconstruction Transformer (LGRT), which integrates multi-granular positional encoding (camera, expression, pose), and Landmark Tracking Loss and Sliced Fusion Loss, enabling incremental reconstruction. Extensive experiments on multiple datasets demonstrate that FastAvatar achieves superior reconstruction quality and rendering fidelity compared to state-of-the-art works.

**Strengths:**

- The proposed LGRT architecture is effective in aggregating multi-view / multiple image cues and aligning variable-length inputs, achieving consistent geometric and appearance coherence across frames.
- The method shows substantial quantitative improvements in PSNR/SSIM/LPIPS, outperforming existing baselines across all view settings (1, 4, 8, and 16 frames).

**Weaknesses:**

- The tracking loss (Eq. 9) includes the term $y$, but its definition is missing. It should explicitly state how ground-truth and predicted landmarks $y_{j,i}$, $\hat{y}_{j,i}$ are obtained.
- Similarly, Eq. 10 introduces $L_{\text{mask}}$, but no definition or explanation of this loss term is provided. A clarification of its role and formulation is necessary.
- The framework claims to support arbitrary input lengths, but experiments are limited to at most 16 views. It is unclear whether the model generalizes beyond 16-frame inputs or whether GPU memory becomes a constraint.
- The paper highlights incrementality as a core advantage; however, quantitative improvements saturate after 4 views (Table 1). The authors should discuss why the performance gain diminishes beyond 4~8 views.
- In Table 2, the ablation only considers fixed-view settings. Since Sliced Fusion Loss is designed for multi-input benefit, it would be informative to show how its impact scales with increasing frame counts (e.g., 1, 4, 8, 16 frames w/ or w/o the Sliced Fusion Loss).

I will reconsider the score when all those concerns are handled well.

**Questions:**

What are typical failure cases observed in reconstruction or animation (e.g., inconsistent geometry, occluded regions, or identity drift) in this work?

---

> ### Author Response · Authors · 2025-11-28
> **Response to Reviewer Fo9q**
>
> **Weakness 1**: During data pre-processing, obtaining the FLAME parameters requires detecting facial landmarks, which serves as ground truth $y$.
>
> **Weakness 2**: It was a typo. Both $L_{tracking}$ and $L_{mask}$ should be included in the loss function. $L_{tracking}$ helps with points registration, and $L_{mask}$ removes redundant points. We have updated the paper with a detailed explanation (page 5 **3DGS Pruning**).
>
> **Weakness 3**: Neither of these is a limitation. FastAvatar can handle hundreds of frames while achieving high-fidelity reconstructions (see **Figure 5** paragraph). Similar to VGGT, which mainly highlights reconstruction from sparse inputs, we primarily report results using 1–16 input frames. This aligns with our training setup, where the number of frames, expressions, and camera poses are randomly sampled, resulting in substantial variation and representing sparse-data reconstruction. Building on this capability, the model can be easily extended to more flexible scenarios. For real-world sequences, adjacent frames are highly correlated, so we sample 16 evenly spaced frames and process the remaining frames through a simple FramePack-inspired 3D convolution module, providing additional context with only a minor increase in computational cost. This design balances efficiency with robust, scalable reconstruction, enabling inference on over 500 frames using a single 48G RTX 4090 GPU.
>
> **Weakness 4**: Because our training data are extremely sparse—formed by randomly sampling motions, video frames, and camera views—FastAvatar naturally learns robust sparse-frame reconstruction. As shown in **Figure 4**, even four input frames already deliver strong results, so adding more frames mostly refines minor details, leading to diminishing gains in both metrics and visual quality.  Nevertheless, richer inputs remain beneficial. As noted above, FastAvatar can obtain further improvements by compressing large numbers of continuous, highly redundant frames into a compact representation.
>
> | Method | PSNR ↑ | SSIM ↑ | LPIPS ↓ | ID ↓ | #GS(K) ↓ | PSNR ↑ | SSIM ↑ | LPIPS ↓ | ID ↓ | #GS(K) ↓ |
> |--------|--------|--------|----------|-------|-----------|--------|--------|----------|-------|-----------|
> |        | **1 frame** | | | | | **4 frames** | | | | |
> | w/o sliced fusion loss | _20.47_ | _0.861_ | **0.123** | 0.158 | 20.7 | _21.69_ | 0.857 | 0.131 | 0.138 | 82.8 |
> | w/o tracking loss | **20.93** | **0.863** | 0.140 | 0.172 | 13.0 | _21.64_ | 0.866 | 0.120 | _0.128_ | 41.2 |
> | w/o global attention | 15.40 | 0.760 | 0.238 | 0.400 | **10.3** | 19.49 | 0.828 | 0.162 | 0.270 | 40.1 |
> | w/o GS fusion | 20.31 | 0.857 | _0.136_ | **0.138** | _12.8_ | 18.94 | 0.838 | 0.157 | 0.185 | **12.5** |
> | w/o GS pruning | 20.32 | 0.857 | 0.137 | _0.144_ | 21.7 | _21.67_ | _0.868_ | _0.112_ | 0.130 | 86.7 |
> | **Ours full** | 20.31 | 0.857 | _0.136_ | 0.148 | _12.8_ | **21.86** | **0.871** | **0.107** | **0.118** | _42.8_ |
>
> ---
>
> | Method | PSNR ↑ | SSIM ↑ | LPIPS ↓ | ID ↓ | #GS(K) ↓ | PSNR ↑ | SSIM ↑ | LPIPS ↓ | ID ↓ | #GS(K) ↓ |
> |--------|--------|--------|----------|-------|-----------|--------|--------|----------|-------|-----------|
> |        | **8 frames** | | | | | **16 frames** | | | | |
> | w/o sliced fusion loss | 21.12 | 0.849 | 0.144 | 0.151 | 165.4 | 20.62 | 0.839 | 0.159 | 0.180 | 330.5 |
> | w/o tracking loss | _21.61_ | 0.867 | 0.119 | _0.124_ | 78.6 | _21.66_ | 0.865 | 0.123 | _0.129_ | 164.2 |
> | w/o global attention | 19.97 | 0.835 | 0.156 | 0.223 | 78.0 | 20.06 | 0.830 | 0.167 | 0.210 | 155.7 |
> | w/o GS fusion | 17.44 | 0.823 | 0.179 | 0.227 | **12.5** | 16.25 | 0.811 | 0.196 | 0.259 | **12.4** |
> | w/o GS pruning | _21.63_ | _0.868_ | _0.110_ | 0.130 | 173.4 | 21.61 | 0.867 | _0.110_ | 0.136 | 346.8 |
> | **Ours full** | **21.95** | **0.871** | **0.103** | **0.118** | _77.0_ | **22.04** | **0.876** | **0.102** | **0.118** | _138.9_ |
>
> ---
>
> **Weakness 5**: We conducted a more comprehensive ablation study covering additional key components and varying numbers of input frames (see Table 2 and **Figure 9**). The results show that when only a single frame is provided, FastAvatar and its variants exhibit similar performance. As the number of input frames increases, mechanisms designed to integrate multi-frame information—such as sliced fusion loss, tracking loss and global attention—begin to take effect. These components improve GS point registration and yield more consistent, high-fidelity reconstructions.
>
> **Questions**: First, our method relies on LBS and FLAME-based encodings to drive 3D head avatar motion, which limits the representation of complex facial muscle dynamics. As a result, the model struggles with wrinkles and eye-gaze, defaults to an average direction. Furthermore, because the Gaussians are anchored to FLAME vertices, the model is unable to represent structures outside the FLAME topology, including the tongue. The Limitation section (**Figure 10**) in the Appendix presents representative failure cases.

---

### Official Review · Reviewer_WZHK · 2025-10-29

**Soundness:** 2
**Presentation:** 2
**Contribution:** 2
**Rating:** 2
**Confidence:** 5

**Summary:**

The paper proposes FastAvatar, a feedforward framework for fast and unified 3D avatar reconstruction from variable-length inputs (single images, monocular sequences, or sparse multi-view frames). The core model, termed LGRT, uses alternating frame/global attention blocks and injects camera pose, expression, and head-pose encodings into image tokens. The model directly predicts 3DGS attributes for avatar reconstruction, while two new losses, including sliced fusion loss and landmark tracking loss, encourage consistent multi-frame fusion and geometric alignment. The paper claims incremental reconstruction capability, where quality improves as more observations are provided.

**Strengths:**

- Practical motivation: Addresses fast, unified avatar reconstruction from variable-length inputs without per-identity optimization.

- Architecture: Extends VGGT with multi-granular encodings (pose, expression, camera) suitable for dynamic faces.

- Loss design: The sliced fusion and landmark tracking losses are reasonable to promote frame consistency and alignment.

- Feedforward inference potentially enables better avatar generation compared to optimization-based methods.

**Weaknesses:**

- Dependence on external camera pose tracking:
Unlike VGGT, which infers relative geometry and camera pose implicitly through attention, FastAvatar requires explicit camera parameters and FLAME-derived head/expression tracking as inputs. This reliance on external preprocessing weakens the method’s claim of being a fully feed-forward, generalizable system. In practice, the need for accurate tracking limits applicability in real-world scenarios (e.g., in-the-wild videos) and reduces the robustness advantage that a unified transformer architecture should ideally provide.
- Methodological inconsistency: Despite criticizing parametric proxies, the method depends heavily on FLAME tracking for pose/expression priors, limiting robustness and novelty.
- Visual evidence lacking:
The qualitative effects aren’t striking. Some reconstructions lack detail and natural dynamics (see video).
All qualitative examples show mainly frontal or near-frontal views, without large-angle or side-view results. This omission makes it difficult to assess whether the method truly captures consistent and complete 3D geometry rather than merely fitting frontal appearances. Given that the paper claims to produce fully animatable 3D avatars, the absence of wide-angle and rotational visualizations significantly weakens the empirical validation of this claim.
- Incremental claim overstated: The model recomputes from all available inputs rather than performing genuine streaming updates.
- Evaluation issues:
   - Modified baselines (e.g., LAM) may yield unfair comparisons.
   - Missing key baselines such as Arc2Avatar, HeadGap, and SynShot.
   - no identity evaluations.
   - The video results are too short.

**Questions:**

- During fusion, how exactly does the Gumbel-Softmax operate? Does it directly set the Gaussian attribute masks for some frames to zero? Please provide more details about its mechanism.
- As the number of input images increases, the performance improvements are limited; interestingly, in Figure 4, the visual results become sharper with more input views, which contradicts the common expectation that multiple views usually cause smoothing. Could this be related to some Gaussian attribute fusion strategy? Where do the authors think this gain originates from?
- Training time? The paper does not report key training details such as the hardware configuration, batch size, training time, or number of GPU hours required for training.
- How many Gaussians are predicted per frame and per fused model? Are they anchored to FLAME vertices or generated freely in canonical space?
- How are LBS weights for Gaussians derived—learned, interpolated from FLAME, or transferred directly?
- What does “incremental” precisely mean? Does the system reuse previous results or recompute from scratch with new inputs?
- Why was LAM modified for multi-frame input, and how was fairness ensured in this comparison?

---

> ### Author Response · Authors · 2025-11-28
> **Response to Reviewer WZHK**
>
> **Weakness 1 & 2**:
> 1. FastAvatar is inherently robust to potential inaccuracies in FLAME tracking, so it does not rely on an ideally precise FLAME input (32-pixel landmark noise reduces PSNR by only 0.5 dB). This robustness is mainly due to the use of photometric supervision during training, which supports the effectiveness of the end-to-end framework. In practice, current FLAME tracking methods are cheap and robust. Minor perturbations have only negligible effect on our feed-forward pipeline’s performance.
> 2. Recent methods such as LAM (SIGGRAPH 2025) and Avat3r (ICCV 2025) also rely on prerequisite processing steps. LAM requires FLAME tracking, while Avat3r depends on DUST3R Sapiens. Despite these dependencies, both are unequivocally presented and recognized as feed-forward 3D avatar reconstruction pipelines. FastAvatar not only follows this paradigm but also unifies it into a more general and scalable formulation that addresses a substantially broader and more challenging range of inputs. This simple, scalable feed-forward paradigm is emerging as a promising blueprint for the next generation of avatar models, offering a unified and extensible framework capable of accommodating ever-growing data diversity, task complexity, and deployment demands.
>
> **Weakness 3**: **Figure 4** illustrates FastAvatar’s ability to produce high-fidelity reconstructions across a variety of expressions, viewpoints, and facial details (e.g., earrings). Additional reconstruction results from more viewpoints are provided Generalization to Wide-Range Viewpoints (Figure 12,13) in the Appendix, which further demonstrate the outstanding performance of FastAvatar.
>
> **Weakness 4 & Question 6**: FastAvatar benefits from additional input frames, improving modeling of diverse and extreme expressions (see Figure 5,11). For streaming data, we employ an overlapping sliding-window strategy; the Gaussian Registration module accurately integrates newly arriving frames, enabling incremental reconstruction. With increased sliding-window coverage, reconstruction quality of the oral cavity—rarely visible in single frames—steadily improves, demonstrating FastAvatar’s incremental modeling capability.
>
> **Weakness 5 & Question 7**:
> 1. We introduce LAM* to demonstrate that existing state-of-the-art feed-forward 3D Avatar reconstruction model lack point registration capability and cannot achieve performance improvements from multiple input frames—a limitation that FastAvatar effectively addresses. Importantly, when only a single frame is provided, LAM* is equivalent to the original LAM; in our implementation, we merely concatenate Gaussians from multiple frames before rendering, without affecting any other components. All LAM* experiments in the paper include 1-view input, ensuring that its inclusion does not create any unfair comparisons;
> 2. Comparisons with Arc2Avatar and SynShot are presented in **Figure 7,8**. Since SynShot and HeadGap do not provide publicly available model weights, we perform a fair comparison using the visual results reported in the papers;
> 3. We included ArcFace-based identity metrics in the comparison tables;
> 4. We added a long video in the supplementary material.
>
> **Question 1**: During modeling, we compute an importance score $m$ for each Gaussian and apply a Gumbel-Softmax relaxation to obtain a binary mask. Pruned primitives (i.e., those assigned a mask value of 0) are excluded during rasterization.
>
> **Question 2**: As shown in Figure 4, concatenating multi-frame information in LAM* leads to noticeable performance degradation. This demonstrates that existing feed-forward 3D head avatar reconstruction methods lack Gaussian registration capability and cannot selectively fuse information across frames, resulting in ghosting artifacts. In contrast, FastAvatar addresses Gaussian registration through Global Attention, Sliced Fusion Loss, and Tracking Loss, enabling multi-frame information to be consolidated into a coordinated representation. Furthermore, Gaussian Pruning removes redundant Gaussians, allowing the model to selectively incorporate fine-grained details as the number of input images increases.
>
> **Question 3**: Details regarding training time are provided in the Reproducibility section of the Appendix.
>
> **Question 4**: The number of Gaussians is reported in Table 2. They are anchored to the FLAME vertices and, without pruning, each frame contains 21,675 points obtained via a single upsampling step, consistent with LAM. When GS pruning is applied, this rule no longer holds: FastAvatar computes an importance score for each Gaussian and performs pruning accordingly.
>
> **Question 5**: The LBS weights for Gaussians are directly transferred from the FLAME model, where each Gaussian inherits the LBS weights of its corresponding canonical vertex without any learning or interpolation.

---

### Official Review · Reviewer_rdbM · 2025-11-01

**Soundness:** 4
**Presentation:** 3
**Contribution:** 3
**Rating:** 8
**Confidence:** 4

**Summary:**

FastAvatar is introduced as a novel feedforward framework for 3D avatar reconstruction, designed to overcome existing challenges in high time complexity, sensitivity to data quality, and inefficient data utilization in contemporary methods.

This single unified model flexibly leverages diverse daily recordings, such as a single image, multi-view observations, or monocular video, to reconstruct a high-quality animatable 3D Gaussian Splatting (3DGS) model typically within seconds. The central component of the framework is the Large Gaussian Reconstruction Transformer (LGRT), which is specifically adapted from the VGGT structure to meet the demanding registration and aggregation needs of 3D avatar tasks. The LGRT incorporates designs including a 3DGS transformer that aggregates multi-frame cues by injecting initial 3D positional prompts, and multi-granular guidance encoding utilizing camera pose, expression coefficients, and head pose to mitigate animation-induced misalignment for variable-length inputs. Crucially, the model utilizes a Landmark Tracking Loss and Sliced Fusion Loss during training, which supervise the combination of frame-wise Gaussian representations to ensure consistency and enhanced aggregation accuracy.

Integrating these contributions, FastAvatar uniquely pioneers incremental 3D avatar reconstruction, allowing the model to continuously ingest new observational data to progressively refine modeling quality while achieving highly competitive quality and speed compared to existing state-of-the-art approaches.

**Strengths:**

- FastAvatar uses a single unified model capable of processing diverse daily recordings, including a single image, multi-view observations, or monocular video. FastAvatar demonstrates greater model flexibility and higher data utilization efficiency compared to other feedforward methods like LAM or Avat3r.

- Sliced Fusion Loss is a key component of the FastAvatar framework. This loss enables the model $G$ to leverage richer information from multiple inputs and to handle an arbitrary number of frames. It supervises the fusion of per-frame Gaussian representations, ensuring cross-frame consistency.

- FastAvatar introduces the capability for incremental 3D avatar reconstruction, a feature currently unattainable by existing approaches. This means the model can continuously ingest new observational data to progressively and reliably refine modeling quality.

**Weaknesses:**

- Although the method handles variable lengths, the practical input size N is explicitly limited. This constraint (max 16 frames) suggests that processing longer videos (which previously required 30 seconds at 25fps, for optimization-based methods) still requires sampling or chunking, potentially limiting true incremental modeling for extended footage.

- The precision of these proxy models (FLAME/3DMM) is known to be sensitive to limitations like representational capacity and data quality, often failing to produce highly accurate proxy 3D models. Although FastAvatar uses this information for alignment, the quality of the final reconstruction is fundamentally tied to the accuracy of these initial estimates. The current ablation study only focuses on the two proposed losses ($L_{track}$  and $L_{sliced}$). It is recommended to include an ablation on the estimated FLAME parameters. Measuring performance degradation would quantitatively assess robustness to real-world FLAME tracking inaccuracies. For example, the authors could test different tracking methods. When tracking is erroneous, does the method propagate these errors, or can the multi-frame image evidence mitigate them?

**Questions:**

The specific contribution of the initial 3D prompt, which distinguishes this approach from other Transformer designs, is not quantitatively demonstrated. The description is unclear: is the LGRT initialization based on standard FLAME vertices, subject-specific identity-shaped FLAME, or another form? Clarification and discussion would be helpful.

---

> ### Author Response · Authors · 2025-11-28
> **Response to Reviewer rdbM**
>
> **Weakness 1**: It should be noted that using 16 input frames is not a limitation. FastAvatar can handle hundreds of frames while maintaining strong performance (see **Figure 5**). In real-world sequences, adjacent frames are highly correlated, so inputting all frames indiscriminately yields only marginal gains while substantially increasing computational cost, as Global Attention is resource-intensive. Accordingly, we use 16 uniformly sampled key frames as the sparse-input representation, while the remaining frames are processed by a FramePack-inspired 3D convolution module that provides additional complementary information with only a minor increase in computation. This design balances efficiency with robust, scalable reconstruction, enabling inference on over 500 frames using a single 48G RTX 4090 GPU. Moreover, existing feed-forward SOTA 3D head avatar reconstruction methods, such as LAM and Avat3r, do not support variable-length input sequences and cannot leverage additional frames to improve performance. To the best of our knowledge, FastAvatar is the first method to achieve this, paving the way for future research in scalable multi-frame 3D avatar reconstruction.
>
> ---
> ### Ablation studies on FLAME tracking
> ---
> | Noise  | L1 ↓       | PSNR ↑   | SSIM ↑  | LPIPS ↓ | Identity ↓ |
> |--------|-----------|---------|--------|---------|------------|
> | 1 px   | _0.0264_  | **22.50** | **0.873** | **0.096** | _0.100_ |
> | 4 px   | 0.0268    | _22.38_  | 0.872  | _0.098_  | **0.099** |
> | 8 px   | 0.0273    | 22.22    | 0.870  | _0.098_  | 0.103     |
> | 16 px  | 0.0277    | 22.10    | 0.869  | 0.099     | 0.102     |
> | 32 px  | 0.0280    | 22.02    | 0.869  | 0.099     | 0.105     |
> | **Ours** | **0.0263** | **22.50** | _0.872_ | **0.096** | **0.099** |
> ---
>
> **Weakness 2**: We further extend our ablation study to evaluate the impact of FLAME tracking on FastAvatar’s performance (see **Table 4** in the Appendix). In our pipeline, facial landmarks are used to initialize FLAME tracking.
> We inject spatial noise to mimic the instability of landmark detection in FLAME tracking pipeline, thereby evaluating the robustness of FastAvatar to the quality of FLAME parameteres.
> The results show that FastAvatar’s reconstruction quality remains largely stable, with only a slight performance decrease even under substantial noise (up to 32 pixels per landmark). This robustness stems from FastAvatar’s reliance on strong photometric and perceptual losses during training, which ensure accurate alignment largely independent of the initialization. These findings indicate that FastAvatar exhibits only weak dependence on FLAME tracking, underscoring its robustness and practicality for real-world deployment.
>
> **Question**: As mentioned in Weakness 2, the contribution of the initial 3D prompt is highlighted in the updated ablation study. The LGRT initialization is based on subject-specific identity-shaped FLAME.

---

### Author Response · Authors · 2025-11-28
**Global Response**

### Summary
In this work, we introduce FastAvatar, a feed-forward framework for high-quality 3D head avatar reconstruction that can process diverse inputs—including a single image, sparse multi-view observations, or monocular video—using a single unified model. Unlike existing feed-forward models such as LAM and Avat3r, which can only utilize a limited number of input frames. While these methods are confined to sparse inputs and consequently often exhibit substantial fidelity issues, FastAvatar can fully leverage complete data, including monocular or multi-view videos. By combining a easily scalable data formulation with a unified feed-forward design, FastAvatar paves the way for a promising new paradigm in general, flexible, and scalable 3D head avatar reconstruction, shape future model development and real-world deployment. In the following global response, we address the reviewers’ main concerns and further demonstrate the feasibility and effectiveness of FastAvatar.
### Responses
**Q1**: Dependence on FLAME tracking and external inputs: Questions regarding the model’s reliance on FLAME tracking for pose and expression priors, its robustness to tracking errors. (Reviewer rdbM Weakness 2, Question 1; Reviewer WZHK Weakness 1, 2)
**A1**: In our pipeline, facial landmarks are used to initialize FLAME tracking. To evaluate FastAvatar’s robustness to tracking imperfections, we inject varying levels of noise into the landmark points. The results (Table 4) show that FastAvatar’s reconstruction quality remains largely stable, with only a slight performance decrease even under substantial noise (up to 32 pixels). This robustness stems from FastAvatar’s reliance on strong photometric and perceptual losses during training, which ensure accurate alignment largely independent of the initialization. These findings indicate that FastAvatar exhibits only weak dependence on FLAME tracking, underscoring its robustness and practicality for real-world deployment.

**Q2**: Incremental reconstruction and input length generalization: Clarifications on what “incremental” means in practice, whether the model truly supports extended sequences beyond the 16-frame range reported, and how performance scales with increasing numbers of frames. (Reviewer rdbM Weakness 1; Reviewer WZHK Weakness 4, Question 2, Question 6; Reviewer Fo9q Weakness 3, 4; Reviewer 8Upx Weakness 4)
**A2**: FastAvatar is not limited to the 16 input frames presented in our main results; these sparse inputs were primarily chosen to demonstrate that high-fidelity reconstruction can already be achieved with few frames. FastAvatar can also handle continuous monocular videos with hundreds of highly similar, overlapping frames. To evaluate this scenario, we conducted an additional experiment (Figure 5) using 500+ input frames. Leveraging a FramePack-inspired compression strategy, we bypass memory constraints while retaining informative cues. Leveraging FastAvatar’s strong Gaussian Registration, we further perform incremental reconstruction on streaming data using overlapping sliding window (Figure 11). The results show that FastAvatar still supports incremental reconstruction—additional frames progressively refine fine details, such as the oral cavity (absent in most frames), and enhance identity consistency—demonstrating both the model’s robustness to longer sequences and its ability to benefit from increased inputs. This addresses the limitation of feed-forward models in fully leveraging complete data—such as entire videos or multi-view inputs—which is crucial for practical modeling.

**Q3**: Component contributions and ablation analyses: Requests for more detailed ablations on key modules (e.g., Gaussian pruning, Global Attention and Gaussian aggregation) to quantify their individual impact. (Reviewer Fo9q Weakness 5; Reviewer 8Upx Weakness 2)
**A3**: To further demonstrate FastAvatar’s robust multi-frame processing capabilities, we provide additional detailed ablation studies (Table 2). These experiments analyze the impact of each component as the number of input frames varies. In particular, besides sliced fusion loss and landmark tracking loss, we examine the contributions of Global Attention, Gaussian Fusion, and Gaussian Pruning, validating their effectiveness. Collectively, these components enable FastAvatar to serve as a fast, feedforward framework for high-quality 3D avatar reconstruction from diverse inputs.

**Q4**: Additional quantitative and qualitative results are needed to further analyze and validate FastAvatar's performance. (Reviewer WZHK Weakness 3, 5; Reviewer Fo9q Question 1; Reviewer 8Upx Weakness 3)
**A4**: **Figure 7,8** presents comparisons between FastAvatar and additional baselines on more diverse data; **Figure 9** provides more detailed visualizations for the updated ablation study; **Figure 12,13** demonstrates FastAvatar’s performance across a wider range of camera angles; and **Figure 10** illustrates some typical failure cases.

---

### Meta-Review · Area_Chair_iynr · 2026-01-02

**Summary:**

The authors have effectively addressed the reviewers' concerns about tracking dependencies, input scalability, and architectural necessity. They showed strong performance under tracking noise and can handle over 500 frames using an incremental sliding-window approach with FramePack. New ablation studies and qualitative results confirmed the effectiveness of the Sliced Fusion Loss and Gaussian Pruning modules. The recommendation is to accept the submission, with encouragement to integrate all feedback into the final version.

Notably, FramePack is a new feature not included in the first submission. The authors should clarify how to "compress all remaining frames into an aggregated token representation," as this would greatly help readers apply the same strategy for scaling.

**Reviewer Concerns:**

- Reviewer rdbM: Concerns are well addressed. The evidence on long-sequence processing with FramePack removes the main doubt.
- Reviewer WZHK: Concerns are well addressed. The empirical evidence on noise robustness is convincing for the concerns on "real-world deployment."
- Reviewer Fo9q: Concerns are well addressed. The authors detailed the ablation in Table 2, isolating contributions from Global Attention, Fusion, and Pruning, and clarified the incremental reconstruction path using overlapping sliding windows.
- Reviewer 8Upx: Concerns are well addressed. The authors added more baselines (Fig 7, 8) and provided the requested ablation studies (Tab 2).

**Reviewer Scores:**

- Reviewer rdbM: Likely to maintain the rating (8).
- Reviewer WZHK: Likely to increase the rating (4 $\rightarrow$ 6).
- Reviewer Fo9q: Likely to raise the rating (6).
- Reviewer 8Upx: Likely to raise the rating (6).

---

### Decision · Program_Chairs · 2026-01-26

Accept (Poster)